



# Volatile Organic Compound fluxes in a subarctic peatland and lake

Roger Seco[1,2], Thomas Holst[1,3], Mikkel Sillesen Matzen[1], Andreas Westergaard-Nielsen[2], Tao Li[1,2], Tihomir Simin[1,2], Joachim Jansen[4,5], Patrick Crill[4,5], Thomas Friborg[2], Janne Rinne[3], Riikka Rinnan[1,2]

[1]Terrestrial Ecology Section, Department of Biology, University of Copenhagen, Copenhagen, Denmark
[2]Center for Permafrost (CENPERM), Department of Geosciences and Natural Resource Management, University of Copenhagen, Copenhagen, Denmark
[3]Department of Physical Geography & Ecosystem Science, Lund University, Lund, Sweden
[4]Department of Geological Sciences, Stockholm University, Stockholm, Sweden
[5]Bolin Centre for Climate Research, Stockholm, Sweden

*Correspondence to*: Roger Seco (email@rogerseco.cat)

## Abstract

Ecosystems exchange climate-relevant trace gases with the atmosphere, including volatile organic compounds (VOCs) that are a small but highly reactive part of the carbon cycle. VOCs have important ecological functions and implications for

atmospheric chemistry and climate. We measured the ecosystem-level surface-atmosphere VOC fluxes using the eddy covariance technique at a shallow subarctic lake and an adjacent graminoid-dominated fen in Northern Sweden during two contrasting periods: the peak growing season (mid July) and the senescent period post-growing season (September-October). In July, the fen was a net source of methanol, acetaldehyde, acetone, DMS, isoprene, and monoterpenes. All of these VOCs showed a diel cycle of emission with maxima around noon and isoprene dominated the fluxes ($93 \pm 22$ µmol m$^{-2}$ day$^{-1}$, mean

$\pm$ SE). Isoprene emission was strongly stimulated by temperature and presented a steeper response to temperature ($Q_{10} = 14.5$) than that typically assumed in biogenic emission models, supporting the high temperature sensitivity of arctic vegetation. In September, net emissions of methanol and isoprene were drastically reduced, while acetaldehyde and acetone were deposited to the fen, with rates of up to $-6.7 \pm 2.8$ µmol m$^{-2}$ day$^{-1}$ for acetaldehyde.

Remarkably, the lake was a sink for acetaldehyde and acetone during both periods, with average fluxes up to $-19 \pm 1.3$ µmol

m$^{-2}$ day$^{-1}$ of acetone in July and up to $-8.5 \pm 2.3$ µmol m$^{-2}$ day$^{-1}$ of acetaldehyde in September. The deposition of both carbonyl compounds correlated with their atmospheric mixing ratios, with deposition velocities of $-0.23 \pm 0.01$ and $-0.68 \pm 0.03$ cm s$^{-1}$ for acetone and acetaldehyde, respectively.

Even though these VOC fluxes represented less than 0.5% and less than 5% of the $CO_2$ and $CH_4$ net carbon ecosystem exchange, respectively, VOCs alter the oxidation capacity of the atmosphere. Thus, understanding the response of their

emissions to climate change is important for accurate prediction of the future climatic conditions in this rapidly warming area of the planet.





## 1 Introduction

Arctic climate is warming twice as fast as the global average (Post et al., 2019). This is due to a number of climate system

feedbacks, including albedo change due to retreating snow cover and sea ice, and the forest cover expansion to the open tundra (Overland et al., 2014; Post et al., 2009). Northern ecosystems are known to exchange climate-relevant trace gases with the atmosphere, not only long-lived greenhouse gases such as carbon dioxide ($CO_2$) or methane ($CH_4$) but also hundreds of different volatile organic compounds (VOCs) that are a highly reactive part of the carbon cycle (Rinnan et al., 2014). Trace gases originate from sources as diverse as soils, peats, vegetation and lakes, and currently several of them show a trend towards

greater emission rates with climate warming (Kramshøj et al., 2019; Lindwall et al., 2016a; Wik et al., 2016). At the same time, the warming-induced expansion of woody shrubs into tundra ecosystems (Myers-Smith and Hik, 2018) could enhance the photosynthetic uptake of $CO_2$ and offset concurrent increases in heterotrophic respiration (Mekonnen et al., 2018). Indeed, $CO_2$ and $CH_4$ have been extensively surveyed in high latitudes due to a potential increase in their atmospheric concentrations and connected climatic effects as permafrost thaws (Natali et al., 2019; Schuur et al., 2015). In contrast, far less research has

been devoted to VOC emissions in these areas. VOCs play essential ecological roles: they can mediate the communication between living organisms and protect plants from biotic and abiotic stresses (Baldwin et al., 2006; Filella et al., 2013; Kessler and Baldwin, 2001; Peñuelas et al., 2005; Pichersky and Gershenzon, 2002; Seco et al., 2011b; Velikova et al., 2005). In addition, biogenic VOCs engage in chemical reactions that substantially modify the oxidation capacity of the atmosphere. For example, VOCs enhance the lifetime of methane by competing for its atmospheric oxidants, promote the formation of

tropospheric ozone, and ultimately contribute to aerosol formation in the atmosphere, which has climatic consequences (Atkinson, 2000; Liu et al., 2016; Seco et al., 2011a; Tunved et al., 2006). Meanwhile, climate change is affecting the fundamental functions of VOCs and altering their emission rates in northern ecosystems, directly via warming and indirectly by inducing changes in vegetation composition (Faubert et al., 2010a; Li et al., 2019; Lindwall et al., 2016a; Valolahti et al., 2015). Moreover, the atmospheric impact of VOCs emitted from natural sources may be comparatively more important in

northern latitudes than in more populated territories due to the relatively low presence of anthropogenic VOC emissions (Paasonen et al., 2013).

VOC studies over the last few years have reported strong increases in arctic and subarctic vegetation emissions in response to moderate warming (Faubert et al., 2010a; Kramshøj et al., 2016; Lindwall et al., 2016a), highlighting the capacity of Arctic plants to respond to increasing temperature. However, these studies have been conducted using enclosure techniques with a

range of unwanted side effects, such as temperature and humidity rise inside the enclosure and interactions with chamber materials (Ortega and Helmig, 2008). Micrometeorological techniques such as eddy covariance (EC) can overcome many of those undesirable side effects and, in addition, provide a more representative ecosystem-level VOC exchange quantification by minimizing potential sampling error when upscaling from a small number of enclosures to ecosystem fluxes. However, instrumental challenges have limited the number of ecosystem-level VOC studies using EC and only two such datasets over

short time periods are available for high latitude ecosystems (Holst et al., 2010; Potosnak et al., 2013). Moreover, there are



few or no air-water direct VOC flux measurements from northern lakes. Arctic latitudes possess one of the highest concentration and area of inland water bodies of our planet (Verpoorter et al., 2014). Despite being acknowledged as important regional and global sources of carbon dioxide and methane (Tranvik et al., 2009; Wik et al., 2016), the role of northern lakes in the VOC budget is largely unexplored. It is therefore vital to assess the lake and vegetation VOC fluxes in these high latitude

areas exposed to large environmental changes to be able to estimate their impacts on the regional carbon cycle, atmospheric chemistry and climate.

We report here ecosystem-level VOC fluxes measured by EC from a subarctic shallow post glacial lake, a common lake form in the Arctic (Wik et al., 2016), and its adjacent fen dominated by tall graminoids. We aimed to identify which compounds were released at significant rates from the lake, which from the fen, during two contrasting periods: the peak of the growing

season (mid July) and the senescent period post-growing season (September-October). Further, we assessed the deposition of compounds to the two ecosystems. In addition, we included here the EC fluxes of carbon dioxide and methane from both fen and lake, which have been discussed elsewhere (Jammet et al., 2015, 2017; Jansen et al., 2019), to provide a more thorough overview of the trace gas fluxes during our study.

## 2 Materials and methods

**2.1 Site description and field campaign outline**

Stordalen Mire is a subarctic palsa mire complex underlain by discontinuous permafrost (Johansson et al., 2006). It is located ca. 10 km east of Abisko in northern Sweden (68º20' N, 19º03' E). Local meteorology has been monitored at the Abisko Scientific Research Station (ANS) with records since 1913 (Callaghan et al., 2010). The average air temperature at ANS during the period 1981-2010 was 0.1 °C and average annual rainfall 332 mm; the mean annual air temperature and precipitation at

Stordalen in 2018 were -0.04 °C and 340 mm, respectively. The Stordalen Mire was not affected by the drought that occurred in large parts of central and northern Europe during 2018 (Buras et al., 2020).

Our measurements took place during the 2018 growing season from a 2.92 meter-tall eddy covariance mast located on the shore of a shallow post glacial lake, Villasjön. The geomorphological setting within the Torneträsk catchment determines a bimodal distribution to the surface wind flow of roughly ESE and WNW (Fig. 1). The lake edge mast is influenced by Villasjön

to the east and by fen to the west. The fen is a permafrost-free, minerotrophic wetland with vegetation dominated by tall graminoids, mainly *Carex rostrata* and *Eriophorum angustifolium* (Palace et al., 2018). The lake is 0.17 km$^2$ in area, the largest of the 27 lakes that constitute the 15 km$^2$ Stordalen catchment (Lundin et al., 2013). It has a maximum depth of 1.5 m and it usually freezes close to the bottom in winter (Jansen et al., 2019). The bidirectional flow pattern throughout the year allows a clear distinction of the surface source or sink influences of the surface boundary layer. Thus, depending on the prevailing wind

direction, the measured VOC flux data were assigned to either the lake or the fen (e.g. Jammet et al., 2015). Flux data originating outside of these wind directions, usually at very low wind speeds, contributed only a minor part of the total fluxes





and were excluded from analysis. The average EC flux footprint used in our study (Fig. 1) was calculated with a two-dimensional model (Kljun et al., 2015), see the supplementary material for further details.

Several major instrumental issues precluded us from obtaining a season-long dataset of VOC fluxes. With the available data,

when the instruments were operational, we present measurements from two distinct periods: the peak of vegetation activity (15-22 July) and the senescent period post-growing season (20 September-13 October). We determined the relative seasonal status of the vegetation at this site based on greenness data captured by an automatic camera installed on the EC mast (see Sect. 2.3 and Fig. S1). For the sake of simplicity, in this article we will refer to the season peak period as July and to the post growing season as September.

Considering the wind direction partitioning and the two periods for which we have data, the maximum number of half-hour EC fluxes in July were 64 and 246 for lake and fen, respectively. In September, we had a maximum of 429 EC fluxes for lake and 619 for fen. However, since some VOC fluxes were discarded according to strict quality assurance procedures recommended for EC measurements (see section 2.2), the actual number of valid half-hour VOC fluxes was smaller for individual chemical species (e.g., for isoprene in July, 22 and 161 valid data points for lake and fen, respectively). VOC fluxes

from the lake for July were only available for some hours of the day (between 03:00 and 14:00 UTC+1).

## 2.2 VOC measurements

Mixing ratios of VOCs were measured with a Proton Transfer Reaction – Time of Flight – Mass Spectrometer (PTR-TOF-MS). This particular model (PTR-TOF 1000ultra, Ionicon Analytik, Innsbruck, Austria) was equipped with an ion funnel at

the end of the drift tube that provided a higher sensitivity, and had a compound (diiodobenzene) added continuously to the air sample that provided a constant signal at high mass-to-charge ratio (m/z) for accurate TOF mass scale calibration. The drift tube was operated at 60 ℃, 550 V, and 2.3 mbar. Multipoint sensitivity calibration was performed at least once a month (including at the end of the July period and before, midway and at the end of the September period) by diluting a blend of several VOCs (e.g., methanol, acetaldehyde, acetone, isoprene, alpha-pinene, benzene, among others, in nitrogen at a VOC

mixing ratio of $1 \times 10^{-6}$ mol mol$^{-1}$ each, manufactured by Ionicon Analytik) into clean nitrogen. The dilution (range 1-40 $\times 10^{-9}$ mol mol$^{-1}$) was achieved with a Liquid Calibration Unit (Ionicon Analytik). The background signal of the PTR-TOF-MS was checked for 1 hour every night by sampling VOC-scrubbed air produced by passing ambient air through a zero air generator (Parker Hannifin 75-83-220, Lancaster NY, USA). The PTR-TOF-MS instrument was sheltered inside a hut located approximately 15 meters from the eddy covariance mast. Air was sampled from the top of the mast, very close to an R3-50

ultrasonic anemometer (Gill Instruments, United Kingdom), at a flow rate of 20 liters per minute through a PFA (perfluoroalkoxy) teflon line (3/8'' OD) inlet that was heated when ambient temperatures were below 20 ℃ to minimize VOC and water condensation onto the inlet walls. The teflon line and its heating wire were inserted into a plastic pipe to shelter them from the environment.





Raw PTR-TOF-MS data were processed with the *PTRwid* software (Holzinger, 2015). PTRwid corrected the mass scale
calibration, and then detected, fitted and quantified ion peaks present in the measured spectrum. In this study, we focus on
several VOCs that were assigned to known protonated ions detected with the PTR-TOF-MS: methanol (m/z 33.03),
acetaldehyde (m/z 45.03), acetone (m/z 59.05), dimethyl sulfide (DMS, m/z 63.03) isoprene (m/z 69.07), and monoterpenes
(m/z 81.07 and 137.13).

Fluxes of VOCs were calculated with the eddy covariance technique using the *InnFLUX* software tool by Striednig et al.
(2020), which was run in Matlab version R2018b (The Mathworks, Natick MA, USA). In short, InnFLUX first detrended the
Reynolds averages of the raw data. Then, for each half hour EC flux calculation, it time-aligned the VOC mixing ratio time
series for each m/z with the vertical wind data from the sonic anemometer by shifting one time series relative to the other until
the absolute maximum covariance between the two time-series was determined. This time alignment also corrected for the
variable time difference between the computer recording the PTR-TOF-MS data and the data logger recording the wind data.
Previously, the wind data had been rotated according to the directional planar fit method, a correction dependent on wind
direction that applies the planar fit method (Wilczak et al., 2001) to each wind direction (1 degree steps) using a rotational
matrix computed with wind data of the ±15 degrees around that wind direction. Calculated fluxes were excluded from further
analysis if turbulence was low ($u^* < 0.15$) or if results of the stationarity test (Foken et al., 2004) were higher than 30%. Of
the total calculated half-hour EC fluxes, those excluded by these conditions represented 34% for isoprene, 40% for
monoterpenes, 43% for acetone and DMS, 44% for methanol, and 62% for acetaldehyde.

Even though VOC measurements were made at 10 Hz, the damping of the turbulence inside the long inlet line and other
possible losses of high frequency contributions to the VOC flux required the application of spectral corrections to the calculated
fluxes. We chose the empirical method proposed by Aubinet et al. (2001), which derives a cospectral transfer function of the
EC system based on the comparison of the covariance of the vertical wind speed with a non-attenuated signal (i.e. the
temperature measured directly by the sonic anemometer) to that with an attenuated signal (i.e. the VOC; Fig. S2). Since the
flux mast setup did not change during the measurement campaign, the spectral correction factor (i.e. number to be multiplied
by the attenuated flux to obtain the corrected flux) was simply a function of the wind speed (Aubinet et al., 2001). The
correction factor determined for isoprene was used for all reported compounds and ranged from 1 to 1.6, and on average was
1.2, which corresponded to a wind speed of 4 m s$^{-1}$. In addition, the EC system response time ($\tau_c$=0.21 s), including the PTR-
TOF-MS and the inlet tubing, was calculated by fitting the same cospectral transfer function to the equation by Horst (1997;
Eq 5 therein).

## 2.3 Ancillary measurements

Simultaneously with VOCs, and at the same mast, we measured EC fluxes of $CH_4$ and $CO_2$. The setup as well as the data
processing procedures are detailed in Jammet et al. (2015, 2017) and Jansen et al. (2019). This EC system used the same Gill
R3-50 sonic anemometer as the PTR-TOF-MS. For the $CH_4$ fluxes, a tube inlet (8 mm ID Synflex), mounted just below the



anemometer, was connected to a closed path cavity ring-down spectrometer (FGGA, Los Gatos Research, San Jose CA, USA). For $CO_2$ and $H_2O$ fluxes we used an open path sensor (LI7500a, LI-COR Biosciences, Lincoln NE, USA) mounted at 2.5m height. Digital data streams from the instruments were sampled at 10 Hz and stored on a CR3000 datalogger (Campbell

Scientific Inc., Logan UT, USA). Raw data processing, flux calculations and spectral corrections were done in EddyPro version 6.2 (open source software hosted by LI-COR).

A second mast was set up ca. 10 m from the bank of the lake equipped with instrumentation for ancillary data. Air temperature and humidity (CS215 probe, Campbell Scientific) were measured at 2 m a.g.l. Incoming and reflected photosynthetic active radiation (PAR) were recorded at 1.5 m a.g.l. using two Li-190 probes (LI-COR Biosciences). All these data were recorded as

10 min-averages and stored using a CR1000 datalogger (Campbell Scientific).

The vegetation greenness was measured throughout the growing season (Fig. S1) for a general assessment of vegetation phenology and growing season dynamics. The greenness was derived as the green chromatic coordinate (GCC; Westergaard-Nielsen et al., 2017), based on images acquired every second hour with an automated Canon G7 X Mark II camera. We visually selected four regions in the images dominated by *Betula pubescens* var. *pumila* L.*, Betula nana* L.*, Salix* spp., and graminoids

(*Carex* spp. and *Eriophorum* spp.), respectively, and averaged the GCC within each region, based on the average of four daily images from approx. 10:00 to 16:00.

We used the vegetation surface temperature, retrieved with an infrared radiometer (SI-111, Apogee Instruments, Logan UT, USA), from the nearby ICOS (Integrated Carbon Observation System) Sweden measurements within the same Stordalen Mire complex (Fig. 1). Water temperature in Villasjön was measured with self-contained loggers (HOBO Water Temp Pro v2, Onset

Computer) at 0.1, 0.3, 0.5 and 1.0m depth, at 5-minute intervals. The loggers were intercalibrated in a well-mixed water tank prior to deployment to achieve a measurement precision of < 0.05 °C.

## 3 Results and discussion

### 3.1 Meteorological conditions

PAR and air temperature were much higher in July than in September (Fig. 2). Maximum hourly average PAR in September was one fifth of that in July (ca. 250 and 1200 µmol m⁻² s⁻¹, respectively). Average air temperature at 2 m height was higher when the wind was blowing from the east, especially in July, with hourly averages ranging from 14 to 28 ℃ (11 to 19 ℃ with wind from the west). In September, hourly average air temperatures ranged between 1 and 4 ℃, and at the end of this period

(second week of October) the first snowfall and the first freezing nights of the season occurred. The water temperature at 10 cm depth did not vary as much in July as the temperature of the air blowing over the lake did, fluctuating between 19.5 and 22 ℃ on average, whereas in September the water temperature closely resembled the air temperature and had minimal variations along the day, staying between 1.9 and 3 ℃ (Fig. 2). In contrast, the vegetation surface temperature did oscillate much more





than the air temperature, ranging between 10 ℃ at midnight and 26 ℃ at noontime in July. In September, the surface

temperature showed a similar diurnal pattern but its hourly average range was limited between -1.2 and 3.3 ℃ (Fig. 2).

### 3.2 Fen VOC fluxes

In July, the fen was a net source of methanol, acetaldehyde, acetone, DMS, isoprene, and monoterpenes (Table 1), with a clear

dominance of isoprene ($93 \pm 22$ µmol m$^{-2}$ day$^{-1}$ on average $\pm$ standard error of the mean). These VOCs showed a typical diurnal

cycle with maximum emission around midday (Fig. 3). In September, fen emissions of methanol, isoprene and monoterpenes

were drastically reduced, reaching mean net emissions of 1 µmol m$^{-2}$ day$^{-1}$ or less, and none of them followed a discernible

diel pattern. Acetaldehyde, acetone, and DMS, in contrast, were deposited to the fen ecosystem with net average rates of -6.7

$\pm 2.8$, $-2.5 \pm 0.3$, and $-0.2 \pm 0.1$ µmol m$^{-2}$ day$^{-1}$, respectively (Table 1). Thus, acetaldehyde and acetone deposition represented

the bulk of VOC flux in the fen after the growing season.

Several biogenic VOC studies in recent years have revealed that isoprene is the major compound in VOC emissions of many

arctic and subarctic ecosystems (Holst et al., 2010; Potosnak et al., 2013; Tiiva et al., 2007), including leaf-level measurements

of two of the dominant sedge species of our fen: *C. rostrata* and *E. angustifolium* (Ekberg et al., 2009). Only two studies,

though, measured the ecosystem-scale emissions with EC, one nearby in the Stordalen Mire fen and the other in an Alaskan

moist acidic tundra (Holst et al., 2010; Potosnak et al., 2013). They published maximum hourly isoprene fluxes that were in

the range of 1 to 5.5 nmol m$^{-2}$ s$^{-1}$ at the peak of the growing season, comparable in magnitude to our July fen measurements

(Fig. 3). A third study that relied on micrometeorology (relaxed eddy accumulation, in this case) was carried out at a boreal

fen in southern Finland dominated by *Sphagnum* spp. mosses. The summer isoprene emissions there were also comparable to

our results, with daily average emissions in the range of 35-88 µmol m$^{-2}$ day$^{-1}$ (Haapanala et al., 2006).

Most of the other published studies derived ecosystem-scale isoprene fluxes with measurement chambers attached to the

ground, enclosing the vegetation and the soil surface together. Pioneering VOC work started in boreal *Sphagnum* fens of

Sweden and Finland with static chambers, finding isoprene fluxes of up to 9 nmol m$^{-2}$ s$^{-1}$, which is in range with our

measurements (Janson et al., 1999; Janson and De Serves, 1998). At a subarctic fen dominated by *Eriophorum* spp. and located

close to our site in the same Stordalen mire complex, Bäckstrand et al. (2008) measured semi-continuously with automatic

chambers and found average total non-methane volatile organic compound (NMVOC) fluxes of 18.5 mgC m$^{-2}$ day$^{-1}$. Assuming

most of the NMVOCs were isoprene, this would equal 309 µmol m$^{-2}$ day$^{-1}$, which is approximately three times our result

(Table 1). Several other push-pull chamber studies have published results from manipulative experiments and, in most cases,

the non-manipulated control plots showed substantially lower isoprene emissions than the EC and chamber studies mentioned

above. For instance, emissions in a Finnish subarctic peatland dominated by the moss *Warnstorfia exannulata* and the sedges

*Eriophorum russeolum* and *Carex limosa* were in general at least one order of magnitude lower than ours but showed great

year to year variation with some higher fluxes measured as well (Faubert et al., 2010b; Tiiva et al., 2007). At a fen in Greenland,

dominated by the graminoids *Carex rariflora* and *E. angustifolium*, isoprene emissions were also at least one order of





magnitude smaller than our results (Lindwall et al., 2016b). At an experimental site in a heath near Abisko, dominated by evergreen and deciduous dwarf shrubs, graminoids and forbs and with soil covered by *Sphagnum warnstorfii*, the average isoprene fluxes were one or two orders of magnitude lower than ours as well (Tiiva et al., 2008; Valolahti et al., 2015).

The instantaneous isoprene emission rate depends on the short-term (seconds to hours) light and temperature conditions (Monson et al., 2012). To assess the relative contribution of light and temperature in controlling the fen isoprene emissions, we used the light ($C_L$) and temperature ($C_T$) activity factors of the G93 empirical leaf-level isoprene emission model (Guenther et al., 1993) and applied it to our ecosystem-level emissions, assuming that the fen vegetation is a single big leaf (e.g. Geron et al., 1997; Seco et al., 2015, 2017). In addition, with the G93 model we could also estimate the standard isoprene emission

potential of the fen (at standard conditions: 1000 µmol m$^{-2}$ s$^{-1}$ of PAR and surface temperature of 30 °C), which was $5.8 \pm 0.13$ nmol m$^{-2}$ s$^{-1}$ (Supplement Fig. S5), almost double that calculated for a boreal *Sphagnum*-dominated fen in southern Finland (Haapanala et al., 2006). The evaluation of the relative contribution of light and temperature to the control of isoprene emissions revealed that, even though light is required for isoprene biosynthesis and subsequent emission, $C_L$ did not play a prominent role in driving the short-term variations of the isoprene fluxes and most of the actual emission regulation was in

response to temperature (Fig. S5). However, our highest isoprene fluxes did not follow the expected $C_T$ temperature relationship and fell above the regression line (Fig. S5), suggesting a stronger response of our highest isoprene fluxes to temperature. The $C_T$ algorithm defines an exponential increase of isoprene emission with temperature until it reaches a maximum at an optimal temperature (35 to 40 °C) and beyond that emission decays when denaturation of enzymes occurs (Guenther et al., 1993). At the relatively low temperatures of the Stordalen Mire, where vegetation surface temperatures exceed

35 °C just on three half-hour periods during our campaign, only the increasing part of the response described by $C_T$ was observable. We thus fitted an exponential relationship between our measured isoprene flux and both the air and surface temperatures and calculated their respective $Q_{10}$ temperature coefficients (Fig. 5). In this case, the $Q_{10}$ coefficient represents the factor by which isoprene emission increases for every 10-degree rise in temperature. The $Q_{10}$ of the G93 $C_T$ activity factor is 3.3, comparable to similar biogenic VOC emission models with $Q_{10}$ values between 3 and 6 (Peñuelas and Staudt, 2010).

These values over 3 are an indication of the synergy, as temperature rises, between the metabolic processes underlying isoprene emissions, namely isoprene synthase activity and availability of its substrate dimethylallyl diphosphate, that each have $Q_{10} \approx$ 2 like biochemical reactions in general (Sharkey and Monson, 2014). For our calculation, we used only emission data points for which light was not a limiting factor (i.e. PAR > 1000 µmol m$^{-2}$ s$^{-1}$) to avoid interference from the correlation between light and temperature when fitting the temperature response. The $Q_{10}$ of isoprene emission in response to vegetation surface

temperature was 14.5 and the $Q_{10}$ in response to air temperature was much higher, 131 (Fig. 5).

Summarizing the results from an experimentally-warmed site, Tang et al. (2016) calculated the isoprene emission temperature response of a subarctic permafrost-free heath ecosystem in Abisko using an Arrhenius-type exponential algorithm, obtaining a $Q_{10}$ coefficient of 10. Kramshøj et al. (2016) performed similar warming experiments at a dry Arctic tundra heath in Greenland, obtaining a $Q_{10}$ of 22. These two $Q_{10}$ coefficients are higher than that of the $C_T$ algorithm ($Q_{10}$ =3.3) and closer to

our result using the vegetation surface temperature ($Q_{10}$ =14.5; red squares in Fig. 5). Both studies used air temperature inside





the chambers for their calculations, which is typically higher than the ambient air temperature, considering that the combined effect of solar radiation and limited air circulation normally heats up the inside of the chambers. Details about where the temperature is measured –temperature that later is used to derive temperature responses– can be important because Arctic vegetation exhibits a large discrepancy between its surface temperature and that of the air. For example, the vegetation surface

was on average 8 ºC, and up to 21 ºC, warmer than the air at a heath in Greenland (Lindwall et al., 2016a), which coincides with our fen temperature readings (Fig. 2, Fig. 5). Indeed, the response of our isoprene emissions to air temperature was even steeper ($Q_{10} = 131$; blue triangles in Fig. 5) than to surface temperature, which could translate into increased modelled isoprene emissions if implemented in models that do not calculate the vegetation temperature but instead use air temperature to drive biogenic VOC emissions. At the same Stordalen wetland as our study, Holst et al. (2010) found a steep temperature response

to air temperatures above 15 ºC, in agreement with our results (Fig. 5). Nevertheless, our high $Q_{10}$ values corroborate that Arctic vegetation can have a stronger temperature sensitivity compared to plants from lower latitudes, which underpinned the most used biogenic emission models (Guenther et al., 2006), as already suggested from previous high-latitude studies (Holst et al., 2010; Kramshøj et al., 2016; Lindwall et al., 2016b, 2016a; Rinnan et al., 2014).

There are few accounts of non-isoprene fluxes from subarctic wetland ecosystems. Monoterpene emissions from chamber

experiments in an Abisko heath showed great variability: sometimes in the range of our fen measurements (we found average hourly maximum of 0.06 nmol m$^{-2}$ s$^{-1}$; Fig. 3) while sometimes one order of magnitude lower (Faubert et al., 2010a; Valolahti et al., 2015). Even lower (three orders of magnitude) were the monoterpene fluxes from a subarctic fen in Finland (Faubert et al., 2010b), while a boreal *Sphagnum* fen in Finland had comparable or higher fluxes (averages between 0.05 and 0.2 nmol m$^{-2}$ s$^{-1}$) than our subarctic fen (Janson et al., 1999).

Regarding methanol, EC fluxes investigated at the Stordalen Mire wetland by Holst et al. (2010) reached a noontime average hourly maximum of 1.3 nmol m$^{-2}$ s$^{-1}$ in early August, which is higher than our 0.2 nmol m$^{-2}$ s$^{-1}$ in July (Fig. 3). They measured net methanol deposition clearly at night, whereas we observed a net zero flux at the end of the day (Fig. 3). Deposition of methanol to vegetation during nighttime has been linked to dissolution into dew droplets because methanol is highly soluble in water (Seco et al., 2007). At our site, the wet surface of the fen could potentially play an important role, but our data do not

show any significant methanol deposition. The net methanol fluxes observed at our subarctic fen in July are lower than most of the published methanol fluxes from a diverse array of ecosystem-scale studies, which also confirmed the widespread importance of methanol deposition (Seco et al., 2007; Wohlfahrt et al., 2015). Acetaldehyde and acetone net emissions in July were also smaller than other published fluxes from terrestrial vegetation (Seco et al., 2007), and in September they were mainly deposited. DMS showed the same behaviour as methanol, with mainly emission in July and deposition after the growing season

(Fig. 3).

All these non-isoprenoid VOCs followed a diffuse relationship with temperature and/or light (Fig. S3), reflecting the complex nature of the controls over their fluxes at the ecosystem level (Seco et al., 2007). DMS can be emitted by plants (Fall et al., 1988; Geng and Mu, 2006; Jardine et al., 2010) and also, driven by temperature, from soils (Staubes et al., 1989; Yang et al., 1996). Methanol is produced during plant leaf expansion (Hüve et al., 2007), while acetaldehyde is produced in flooded roots





(Fall, 2003), which is potentially an important source in a waterlogged fen for species not adapted to this growth condition, like shrubs. The graminoids, in contrast, transport air down to their roots through a specialized tissue in their leaves and stems, and do not suffer from anoxia (Schütz et al., 1991). The exchange of acetone, acetaldehyde and methanol between plants and the atmosphere is controlled by the stomatal conductance due to their water solubility and, furthermore, their atmospheric mixing ratios can have influence on the fluxes to some extent (Filella et al., 2009; Jardine et al., 2008; Niinemets and

Reichstein, 2003; Seco et al., 2007). In addition, the peat and its microbial communities are also a potential source and sink for many volatiles that can be exchanged between the soil and the atmosphere from many biogeochemical processes (Albers et al., 2018; Kramshøj et al., 2018; Woodcroft et al., 2018).

Lastly, our analytical system did not capture any sesquiterpene fluxes. We know from chamber measurements that vegetation present at the Stordalen Mire emits sesquiterpenes. For example, the mountain birch (*B. pubescens* var. *pumila*), which covers

an area of almost 600 000 ha in the Scandinavian subarctic, can emit important amounts of sesquiterpenes (Haapanala et al., 2009). Other experiments in nearby Abisko heaths, mentioned above, documented sesquiterpene flux rates similar to those of monoterpenes measured in the same study (Faubert et al., 2010a; Valolahti et al., 2015). Most recently, sesquiterpenes have been detected as the main terpenoid emissions after isoprene at a subarctic wetland in Finland (Hellén et al., 2020). One obvious reason for our lack of sesquiterpene signal is that our flux footprint did not include a significant amount of high-emitting

species since, for example, most of the nearby birch patches were outside of it. Equally important is the fact that, due to their high reactivity, sesquiterpenes are lost through fast chemical reactions and through interactions with our long inlet tubing wall, so presumably they never made it to our PTR-TOF-MS to be detected. Therefore, the measurement of ecosystem-wide fluxes of sesquiterpenes with EC remains a challenge for future field campaigns.

**3.3 Lake VOC fluxes**

The number of available lake half-hour fluxes in July was low (n≈20) due to technical issues, wind direction partitioning, and fluxes discarded by EC quality assurance criteria. These reasons and the consequent lack of data for half of the hours of the day (Fig. 3) justify that we consider these results exploratory. In particular, compounds such as DMS and monoterpenes had mean daily fluxes dominated by one or two hourly average data points that were not actually hourly averages, since they were

based on only one measurement during that hour (i.e. data points without shading in Fig. 3). However, there are so few observations of lake VOC fluxes that it is important that we document the sparse data we have.

VOC fluxes assigned to the lake wind direction could potentially have some influence from the vegetation of the island in the lake, because under certain conditions the edge of the EC tower footprint reached that far. A previous study at this site described $CO_2$ uptake during the day from the lake EC measurements in summertime (Jammet et al., 2017). Such uptake was later

interpreted as a result of the photosynthetic activity on the island, since water sampling indicated the surface water of lake Villasjön was consistently supersaturated with respect to atmospheric $CO_2$ (Jansen et al., 2019). During our weeklong July measurements, we only observed lake $CO_2$ uptake in the early morning (Fig. 4) when VOC emissions were small (Fig. 3).




Hence, we assume that the vegetation of the island did not substantially bias our lake VOC fluxes although some influence cannot be entirely ruled out.

The most striking feature of the lake ecosystem is that it was a sink for acetaldehyde and acetone in both studied periods (Fig. 3, Table 1). Similar to the fen, the deposition of these compounds accounted for most of the VOC flux in the lake in September. Acetone deposition peaked in July with an average of -19 ± 1.3 µmol m$^{-2}$ day$^{-1}$, while that of acetaldehyde was highest in September with -8.5 ± 2.3 µmol m$^{-2}$ day$^{-1}$ (Table 1). There was a correlation of the acetaldehyde and acetone deposition rates with their corresponding atmospheric mixing ratios (Fig. 6), with increasing deposition at higher mixing ratios, resulting in

average deposition velocities of -0.23 ± 0.01 and -0.68 ± 0.03 cm s$^{-1}$ for acetone and acetaldehyde, respectively. Air temperature also influenced the flux of these two carbonyl VOCs. Acetone deposition was more intense at higher air temperatures, in July, when its mixing ratios were also higher (Fig. 6). In contrast to acetone, acetaldehyde did not present a clear relationship with air temperature but its strongest deposition rates occurred at air temperatures below 3 ºC (Fig. 6). The high water solubility of these short-chain oxygenated VOCs helps their deposition from the air to the water, and may partly

explain the correlation of the deposition rate with their atmospheric mixing ratios (Fig. 6).

   These carbonyl compounds have not only natural sources such as emission from vegetation and soil: they also originate from human activities and from atmospheric degradation of other precursor VOCs (Seco et al., 2007). The limited reactivity of acetone in the troposphere makes it relatively long-lived, typically up to 15 days (Singh et al., 2004), which means that the deposited acetone can be advected from far away (Patokoski et al., 2015). We have not found studies in the literature on air-

water fluxes of acetaldehyde or acetone in freshwater environments, only a few in marine environments. Yet those marine studies have contradictory results on whether the ocean is a net sink or a source of acetone (Fischer et al., 2012), suggesting a location-dependent behaviour where tropical and productive areas are a net source while high latitude oligotrophic oceans are either in an air-water equilibrium (i.e. zero net flux) or act as net sinks of acetone (Beale et al., 2013, 2015; Lawson et al., 2020; Marandino, 2005; Schlundt et al., 2017; Taddei et al., 2009; Tanimoto et al., 2014). Villasjön, as an oligotrophic high

latitude lake, would fit in that conceptual framework as a sink of acetone, in agreement with our observations. The direction of the acetaldehyde flux in seawater has been reported to vary along the year during an annual study in UK shelf waters (Beale et al., 2015) and also to be mostly emission during short-term measurements in a Norwegian fjord mesocosm experiment (Sinha et al., 2007). Further, the lake sink of acetaldehyde and acetone detected with our measurements in the snow-free season may be reversed to a source once the valley is covered in snow, as release of acetaldehyde, acetone and other carbonyl

compounds from snow has been documented (Couch et al., 2000).

   Interestingly, even though methanol is more soluble in water than acetone or acetaldehyde (Sander, 2015), its deposition to the lake did not reach the intensity displayed by the two carbonyl compounds (Fig. 3). Instead, average methanol fluxes showed both net deposition and emission along the day during both seasons. Furthermore, in July the overall methanol flux resulted in a net release from the lake (1.8 ± 2.1 µmol m$^{-2}$ day$^{-1}$; Table 1). Again, given the dearth of published observations, we can only

compare our methanol fluxes to marine studies. In contrast to acetone and acetaldehyde and their variable flux direction,



methanol has been reported to be consistently deposited to the ocean surface, where it could represent a supply of energy and carbon for marine microbes (Beale et al., 2015; Sinha et al., 2007; Yang et al., 2013).

Like the fen, the lake also emitted isoprene in July, although without a visible diel pattern given the incomplete dataset (Fig. 3). Nevertheless, our available data showed maximum hourly average net emissions of 1 nmol m$^{-2}$ s$^{-1}$, being the daily average

net rate of $0.24 \pm 0.12$ nmol m$^{-2}$ s$^{-1}$ (equivalent to $20 \pm 10$ µmol m$^{-2}$ day$^{-1}$; Table 1). These numbers are two to three orders of magnitude higher than isoprene emissions calculated at the large temperate oligotrophic lake Constance (Germany) in the month of July, with maximum hourly average emission rates of 0.004 nmol m$^{-2}$ s$^{-1}$ (Steinke et al., 2018). Based on the data from lake Constance, Steinke et al. (2018) suggested that Arctic lakes could rival terrestrial vegetation emissions in these zones where lake areal coverage is high and terrestrial isoprene sources are small. Our numbers do not fully support that suggestion

for the peak of the season at our site, since the fen net emission was roughly 4.5 times that of the lake (Table 1), albeit it may hold in zones with a ratio of lake to vegetation coverage over five. For instance, the Stordalen catchment has 4.5% of lake coverage and 3.9% of fen coverage (Lundin et al., 2016), so the lake to vegetation ratio is 1.2, and much lower if we include other types of vegetation. Still, their suggestion may be valid for other periods. For example, in our case during the senescent period in September, even though the flux magnitudes were much smaller than in July, the lake average isoprene emission was

double that of the fen ($0.6 \pm 0.4$ and $0.3 \pm 0.3$ µmol m$^{-2}$ day$^{-1}$, respectively; Table 1). We found no other report of isoprene fluxes from lakes, despite the likely existence of many sources analogous to those known in seawater such as phytoplankton, seaweeds, bacteria, and cyanobacteria (Broadgate et al., 2004; Exton et al., 2013; Fall and Copley, 2000; Shaw et al., 2003, 2010). A number of available publications suggest that ocean waters are sources of isoprene to the atmosphere at rates comparable to those calculated for Lake Constance, i.e. two orders of magnitude lower than ours (Broadgate et al., 1997;

Kameyama et al., 2014; Li et al., 2017; Sinha et al., 2007).

DMS is a commonly studied marine trace gas because of its role in aerosol and cloud nucleation chemistry (Carpenter et al., 2012) but there are far fewer observations in freshwater environments. As far as we know, no EC measurements of DMS from lakes exist, so the few published studies that report a DMS flux employed alternative techniques to calculate the fluxes, for example using the DMS concentration difference between water and air with an air-water transfer model to calculate the fluxes.

DMS emissions calculated by Steinke et al. (2018) for the 252 meter-deep lake Constance were, as for isoprene, two orders of magnitude smaller (maximum hourly average emission rates of 0.003 nmol m$^{-2}$ s$^{-1}$) than our July fluxes. A study in Canadian boreal lakes estimated DMS emissions up to a few µmol m$^{-2}$ day$^{-1}$ for shallow lakes, which is on the lower range of the July fluxes in the shallow Villasjön. That study also noted that emissions from deeper lakes were smaller than from shallow or medium-depth lakes (Sharma et al., 1999), while a similar study in the same geographical area found average DMS fluxes of

around 1 µmol m$^{-2}$ day$^{-1}$ from lakes ranging in depths from 1.5 to 20 m but with a 5 meter-deep lake showing much higher emissions of up to 4 µmol m$^{-2}$ day$^{-1}$ on average (Richards et al., 1991). Other authors measured DMS concentrations in a stratified lake in North America, at different depths down to 13 m, and concluded that DMS fluxes to the atmosphere must have been insignificant given that DMS was not present in surface and near surface water (Hu et al., 2007). Another study took a different approach and utilized the phytoplankton biomass and its content of DMS precursors in lake Kinneret (Israel)





to estimate an average DMS emission of 3.3 µmol m$^{-2}$ day$^{-1}$ (Ginzburg et al., 1998), similar to our July average of 4.7 ± 3.1

µmol m$^{-2}$ day$^{-1}$ (Table 1). In contrast, DMS fluxes from the ocean have been directly measured by EC in different places around

the globe, with reported emissions as high as 97 µmol m$^{-2}$ day$^{-1}$ (Bell et al., 2013; Marandino et al., 2007, 2009; Smith et al.,

2018), even up to 300 µmol m$^{-2}$ day$^{-1}$ during an unicellullar phytoplankton bloom (Marandino et al., 2008), but in many cases

with average emissions in the same range as our lake July average flux (Huebert et al., 2004; Tanimoto et al., 2014; Yang et

al., 2011b, 2011a).

### 3.4 CO$_2$, CH$_4$, and H$_2$O fluxes

The fluxes of CO$_2$ and CH$_4$ (as well as H$_2$O) were not the focus of this study and, moreover, their temporal patterns and

environmental drivers over several years at the same site have been examined in detail elsewhere (Jammet et al., 2015, 2017;

Jansen et al., 2019, 2020). Here, we mainly included them to contextualize the VOC fluxes and thus provide a broader overview

of the trace gas exchange of our fen and Villasjön during our two measurement periods. Furthermore, as in the case of the July

VOC fluxes, the limited data availability from the lake in July (36 and 51 half-hourly fluxes for CO$_2$ and CH$_4$, respectively)

advises to consider the presented lake trace gas exchanges with prudence.

Net molar fluxes of CO$_2$, CH$_4$, and H$_2$O were at least two, and up to seven, orders of magnitude higher than the VOC fluxes

(Table 1). Water vapour and CH$_4$ showed net average daily emission in both lake and fen and during both periods, while CO$_2$

showed net uptake in July and net release in September, in both lake and fen (Table 1).

Uptake of CO$_2$ and evapotranspiration in the fen followed a well-defined diel cycle likely due to the physiological activity of

the vegetated surface, with maxima around noontime, notably in July (Fig. 4). In September, a similar pattern was apparent in

the fen's diel cycles (Fig. 4), but the magnitude of the daytime fluxes was much smaller. It was so much smaller that the weaker

CO$_2$ uptake during daylight hours did not compensate for the CO$_2$ release during the rest of the day, resulting in a 24-hour

aggregate mean flux that represented a net release of CO$_2$ to the atmosphere from the fen (Table 1). CH$_4$ emissions from the

fen in July were on average 3.5 times higher than in September (Table 1), and their diel emission cycle showed an overall flat

pattern in both periods (Fig. 4).

The lake was a net sink of CO$_2$ in July, especially due to stronger uptake during the early hours of the day, and a net source in

September, during which there was no diel cycle (Table 1, Fig. 4). Evaporation from the lake was approximately 10-fold higher

in July than in September on a 24-hour basis, and compared to the evapotranspiration from the fen, it was higher in July as

well (Table 1, Fig. 4). Daily CH$_4$ emissions from the lake were smaller than from the fen during both periods (Table 1).

A comparison of the VOC carbon fluxes with the fluxes in the form of CO$_2$ and CH$_4$ (Table 1) reveals that the average net

VOC emission of the fen in July, summing the six VOC species reported in this manuscript, represented 0.16 % of the fen net

carbon uptake as CO$_2$ (of which isoprene alone was 0.15 %) and 4.2 % of the net carbon release as CH$_4$ (isoprene alone, 3.9

%). In September, the absolute VOC net carbon flux at the fen, i.e. the total net amount of carbon exchanged in the form of



VOCs, including both the VOCs with net emission and those with net uptake, added up 0.06 % of the net $CO_2$ carbon and 0.76 % of the net $CH_4$ carbon emitted from the fen.

The same comparison for the lake (Table 1) shows that the aggregate absolute VOC net exchange amounts to 0.32 % of the
$CO_2$ and 3.2 % of the $CH_4$ net carbon fluxes of the lake. In September, the absolute VOC net carbon flux in the lake was equivalent to 0.13 % of the net $CO_2$ carbon release flux and 3.8 % of the net $CH_4$ carbon emission flux.

**4 Concluding remarks**

Here we presented an eddy-covariance dataset measured from two distinct common subarctic landscape types: a permafrost-free fen and a shallow post-glacial lake. Isoprene dominated by far the VOC fluxes from the fen at the peak of the season,
while after the growing season the fen was characterized by deposition of acetaldehyde and acetone (Fig. 3, Table 1). Furthermore, the isoprene emissions from the fen in July were strongly stimulated by temperature and, in agreement with previous arctic and subarctic VOC measurements (Holst et al., 2010; Tang et al., 2016), exhibited a higher temperature sensitivity ($Q_{10}$ = 14.5) than described by the temperature response curves typically used in biogenic emission models ($3 \leq Q_{10} \leq 6$), which are based on measurements made in lower latitudes (Guenther et al., 2006).
Our lake VOC fluxes can be considered exploratory due to the low amount of data available. Despite this, they are valuable given the lack of observations of freshwater fluxes. We showed that the lake was a sink of acetone and acetaldehyde in both July and September with average deposition velocities of $-0.23 \pm 0.01$ and $-0.68 \pm 0.03$ cm s$^{-1}$ for acetone and acetaldehyde, respectively (Fig. 3, Fig. 6, Table 1).

The carbon exchanged as VOC net fluxes from both fen and lake constituted less than 0.5% and less than 5% of the $CO_2$ and
$CH_4$ net carbon ecosystem exchange, respectively. These low proportions are probably one of the reasons, together with technical and logistical challenges (Rinne et al., 2016), of the limited amount of existing VOC studies in lakes or high latitude ecosystems. However, technological advances are gradually removing practical obstacles and, in addition, growing concern about climate change repercussions warrants more research in this rapidly warming area of the world, especially given the importance of VOCs as precursors for aerosols (Paasonen et al., 2013; Svenningsson et al., 2008). $CO_2$ and $CH_4$ fluxes are
already under intense investigation to quantify the strength of their sinks and sources (e.g. Jeong et al., 2018; Oh et al., 2020). Recently, arctic VOCs have received increased attention (e.g. Kramshøj et al., 2016, 2018, 2019) and this study is another contribution towards the understanding of VOC budgets in northern wetlands and inland waters.





## Author contributions

RS, TH, MSM, AWN, TL, TS, and JJ performed measurements and contributed data. RR conceptualized and supervised the study and acquired funding to support this research. RS analysed the data and wrote the original draft. All authors contributed to manuscript writing and revision, and read and approved the submitted version.

## Competing interests

The authors declare that they have no conflict of interest.

## Acknowledgements


This work was supported by the European Research Council (ERC) under the European Union's Horizon 2020 research and innovation programme (grant agreement No 771012), the Independent Research Fund Denmark | Natural Sciences, the Swedish Research Council (VR) (ref. 2013-5562), the European Commission under the Seventh Framework Programme project PAGE21 (ref. 282700) and by the Danish National Research Foundation (CENPERM DNRF100).

We are grateful to ICOS Sweden and the Abisko Scientific Research Station for providing excellent logistics for the work. ICOS Sweden is co-funded by the Swedish Research Council.

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





**Table 1: Average (± standard error of the mean) net exchange rates (µmol m$^{-2}$ day$^{-1}$) of VOCs, CO$_2$, H$_2$O and CH$_4$. Negative values represent net uptake by the ecosystem, and positive values represent net emission.**


| Compound | LAKE (µmol m$^{-2}$ day$^{-1}$) | | FEN (µmol m$^{-2}$ day$^{-1}$) | |
|---|---|---|---|---|
| | July* | September | July | September |
| methanol | 1.8 (±2.1) | -0.2 (±0.3) | 7.7 (±1.1) | 1.1 (±0.3) |
| acetaldehyde | -3.8 (±2.4) | -8.5 (±2.3) | 2.6 (±0.9) | -6.7 (±2.8) |
| acetone | -19 (±1.3) | -4.4 (±0.2) | 1.9 (±1) | -2.5 (±0.3) |
| isoprene | 20 (±10) | 0.6 (±0.4) | 93 (±22) | 0.3 (±0.3) |
| monoterpenes | 2.8 (±2.2) | 0.3 (±0.1) | 2.2 (±0.3) | 0.3 (±0.04) |
| DMS | 4.7 (±3.1) | -0.2 (±0.1) | 1.1 (±0.4) | -0.2 (±0.1) |
| CO$_2$ | -6.3E+04 (±3.7E+04) | 2.7E+04 (±9.8E+03) | -3.1E+05 (±7.3E+04) | 4.6E+04 (±1.2E+04) |
| H$_2$O | 3E+08 (±2.7E+07) | 3.2E+07 (±3.4E+06) | 1.2E+08 (±2E+07) | 4.6E+07 (±4.7E+06) |
| CH$_4$ | 6,400 (±1,400) | 950 (±110) | 1.2E+04 (±2.8E+02) | 3,500 (±90) |

*based on a limited dataset from 03:00 to 14:00 hours (UTC+1)




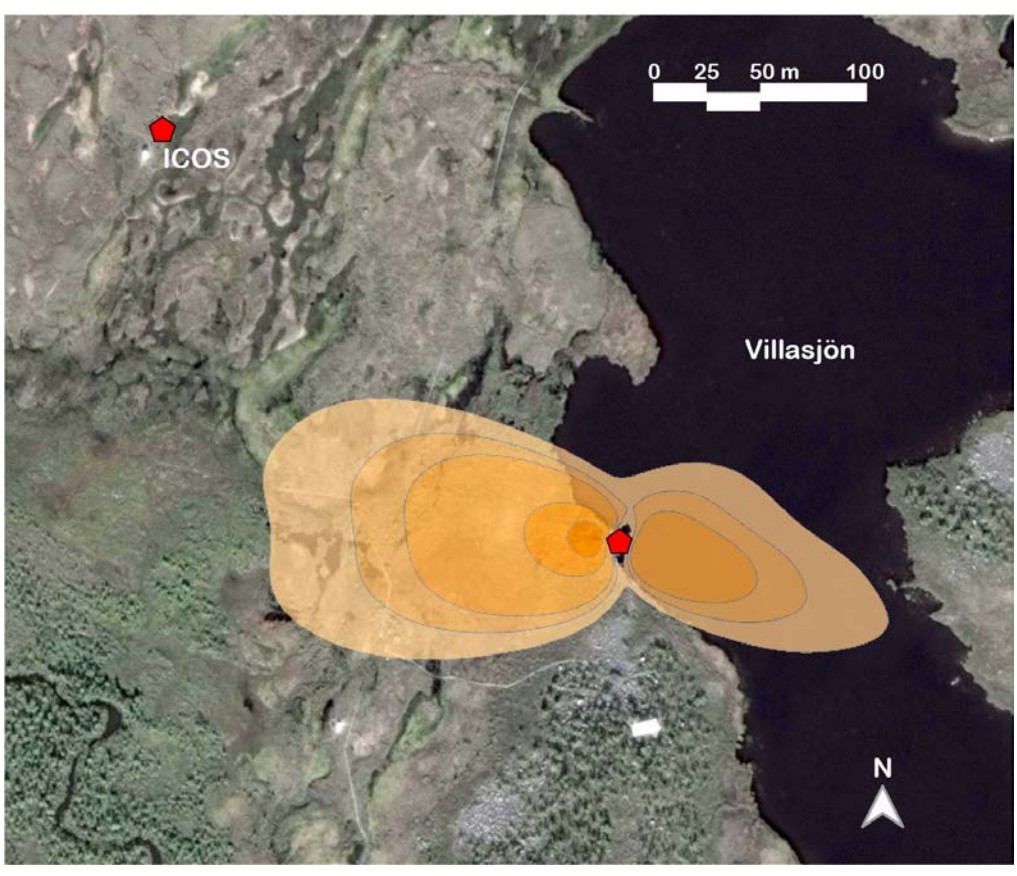

**Figure 1: Map of the Stordalen Mire study area, showing our EC tower by the shore of Villasjön, and the nearby ICOS station location that provided the vegetation surface temperature data. The shaded area represents the combined fen and lake footprint for the July EC measurements, at flux contribution intervals of 85%, 80%, 75%, 50% and 25%. The base map image is © Google Earth (image provided by DigitalGlobe).**


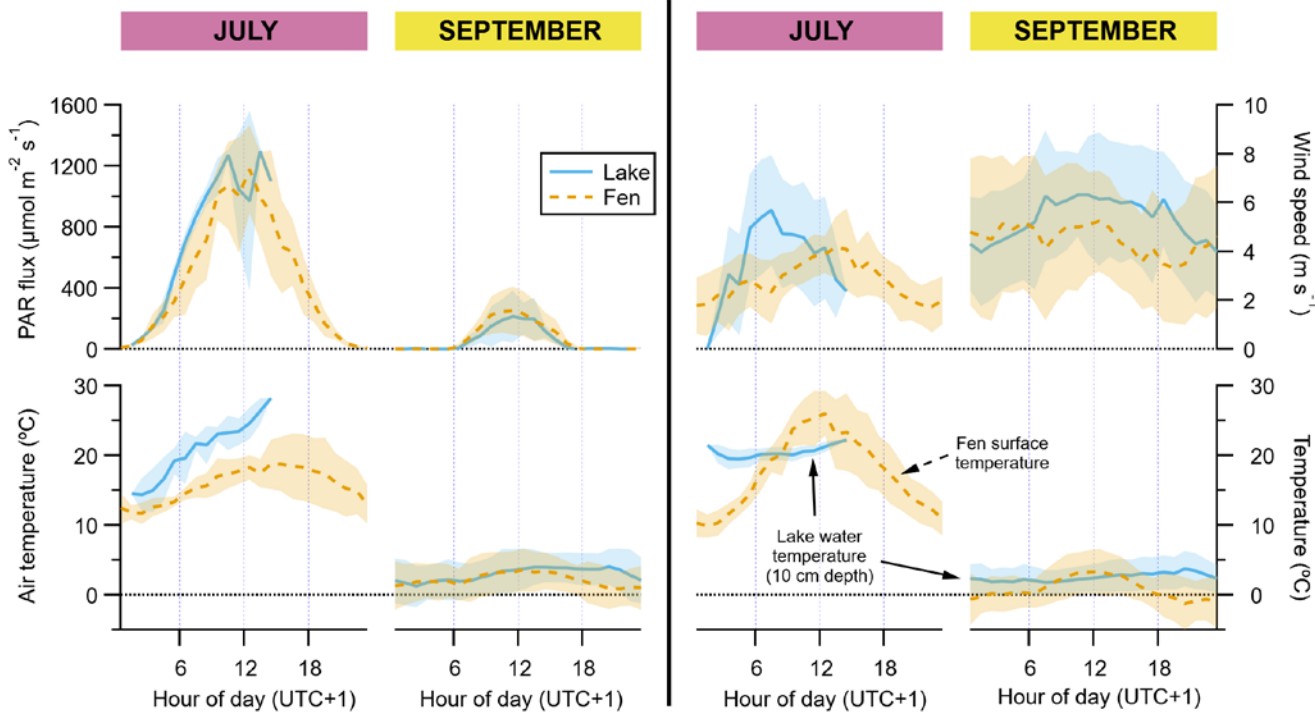


**Figure 2: Diel cycles of hourly averages of meteorological data, for July and September and for both lake (blue solid lines) and fen (dashed orange lines). Peak of growing season (July) is on the left panels of each graph pair, and the post growing season (September) is on the right panels, as indicated on top of the panels. Shaded areas represent ± 1 standard deviation. Each vertical axis has a**
**different scaling, but all of them feature a horizontal dotted line showing where the value zero is located.**




**Figure 3: Diel cycles of hourly averages of VOC fluxes, for July and September and for both lake (blue solid lines) and fen (dashed orange lines). Peak of growing season (July) is on the left panels of each graph pair, and the post growing season (September) is on the right panels, as indicated on top of the panels. Shaded areas represent ± 1 standard deviation. Each vertical axis has a different scaling, but all of them feature a horizontal dotted line showing where the value zero is located.**


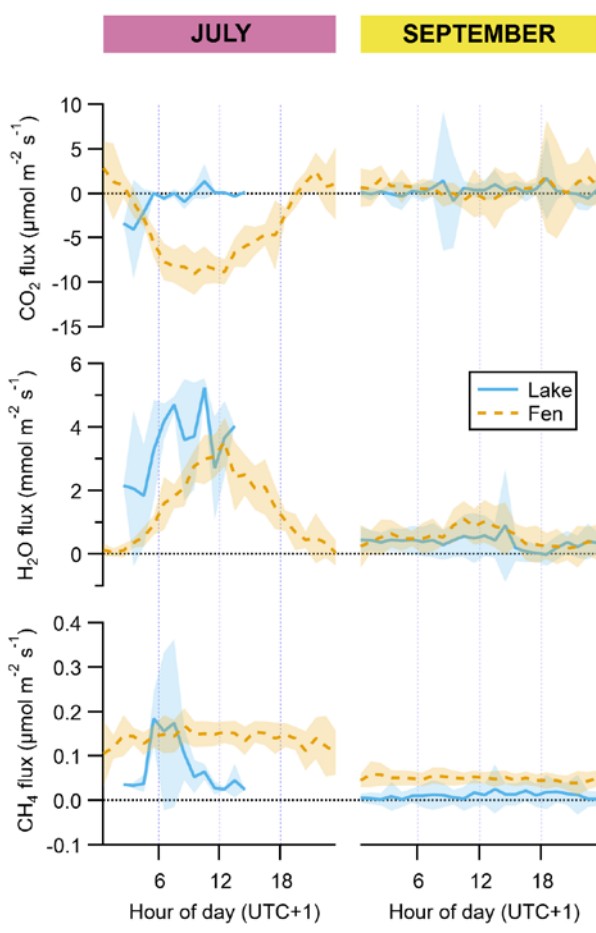

**Figure 4: Diel cycles of hourly averages of fluxes of $CO_2$, $H_2O$ and $CH_4$, for July and September and for both lake (blue solid lines) and fen (dashed orange lines). Peak of growing season (July) is on the left panels of each graph pair, and the post growing season (September) is on the right panels, as indicated on top of the panels. Shaded areas represent ± 1 standard deviation. Each vertical axis has a different scaling, but all of them feature a horizontal dotted line showing where the value zero is located.**







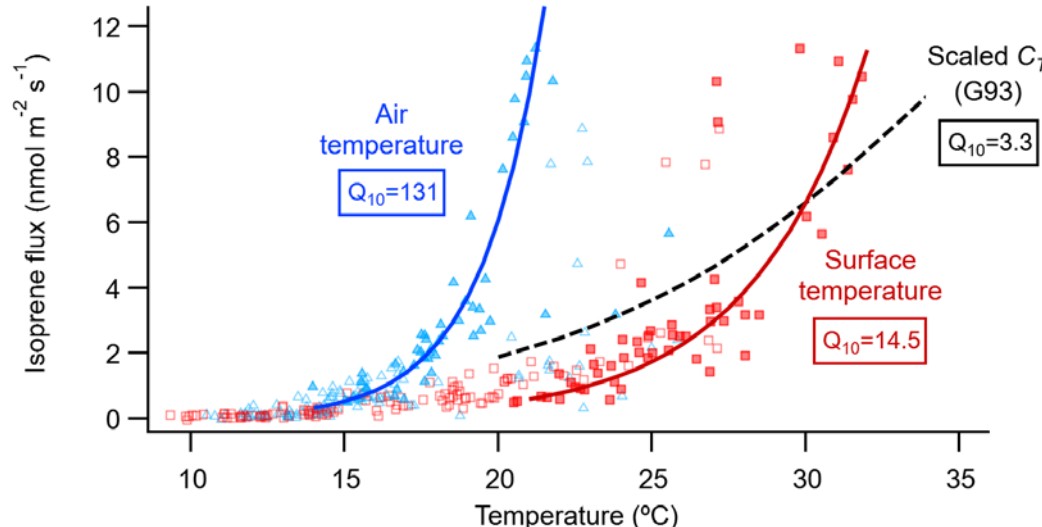

**Figure 5: Relationship of the isoprene flux from the fen with the air temperature (blue triangles) measured at 2 m height on the EC mast, and with the vegetation surface temperature (red squares) measured at the nearby ICOS station. The data points shown are all the July 30-minute fluxes that passed the EC quality criteria (n=161; open and closed symbols). Solid lines show the exponential**

**equation $F = F_0 \cdot Q_{10}^{(T-T_0)/10}$, where $F_0$ is the isoprene flux rate at temperature $T_0$ (=0 ºC), $F$ is the flux rate at temperature $T$ (ºC), and $Q_{10}$ is the temperature coefficient. $F_0$ and $Q_{10}$ were calculated by fitting the data to the linear equation $\log(F) = T \cdot \log(Q_{10})/10 + a_0$, where $a_0$ is the intercept at T=0 ºC. For the fit, only fluxes not limited by light (when PAR was above 1000 µmol m$^{-2}$ s$^{-1}$; n=52; closed symbols) were binned into 1-degree bins (not shown). Then the average fluxes of the bins, excluding bins containing a single flux value, were used to perform an orthogonal distance regression weighed by the standard deviation of each**

**bin average. As a reference, the dashed black line shows the relationship with leaf temperature of the temperature activity factor ($C_T$) of the G93 model (Guenther et al., 1993), scaled to coincide with the surface temperature fit at 30 ºC.**



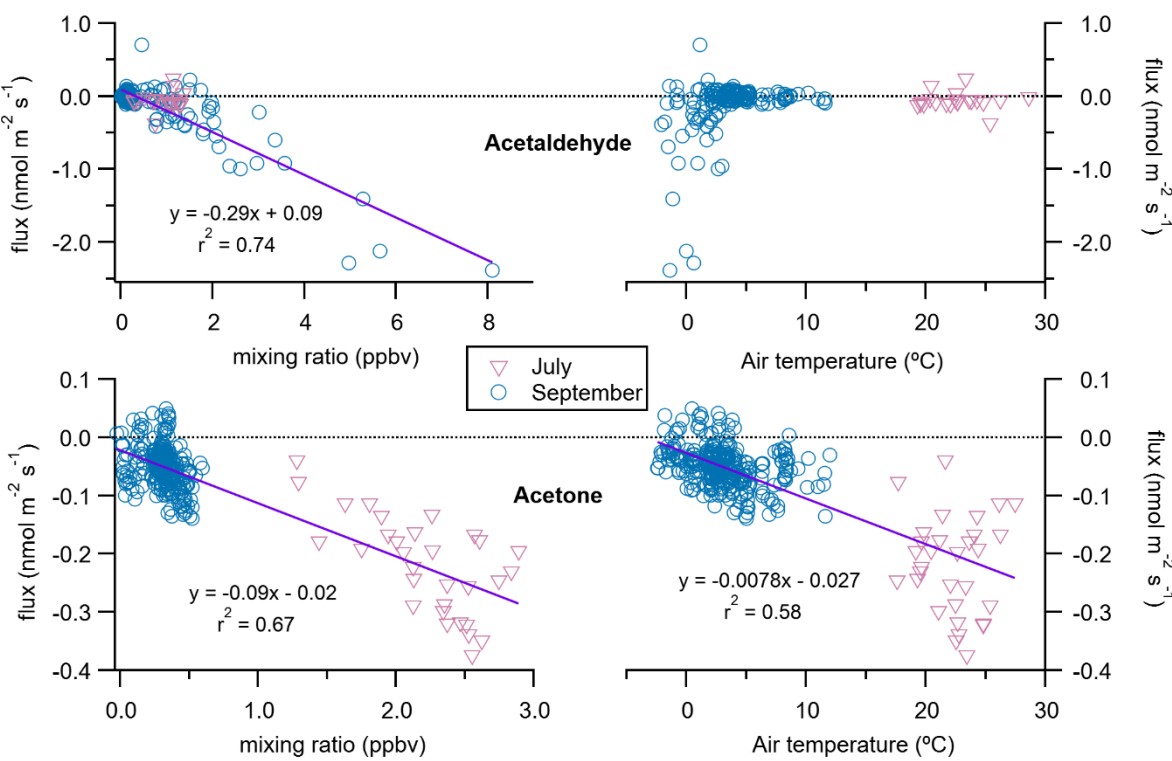


**Figure 6: Relationship of lake fluxes of acetaldehyde (top panels) and acetone (bottom panels) with their respective atmospheric mixing ratios (left panels) and the air temperature (right panels). Each solid line and corresponding equation represent an orthogonal distance regression to all 30-minute flux data points for both July and September (n=191 for acetaldehyde and n=338 for acetone).**
