# Peer review of "Volatile Organic Compound fluxes in a subarctic peatland and lake"

_Atmospheric Chemistry and Physics, 2020_

## Referee Comment (RC1) · Juho Aalto (Referee) · 14 Aug 2020

Seco et al. provide high-quality measurement data representing very relevant study object, VOC emissions from subarctic peatland and lake. Given how rare such datasets are, the results are obviously worth reporting, though the shortness and temporal limitedness of the data set apparent limitations concerning the data analysis and conclusions. These limitations should not be considered as weaknesses of the manuscript – they are more common features that should be taken into account, than any reason for rejection.

Overall, the manuscript fulfills the review criteria of ACP. The scientific question is well within the scope of ACP, both the data and conclusions are mainly valid, as well as

the measurement and data analysis methods. The manuscript itself is mostly well structured and clearly expressed. There are some minor issues in the manuscript that may need to be reconsidered; those issues will be listed in my detailed comments.

Within the following section, I'll go through my main concerns regarding the data analysis and conclusions. Please note that many of my comments are more suggestions and questions aiming to clarify the results, than sound challenging based on solid scientific evidence. Chances are that many of the aspects pointed out by me have already been taken into consideration during the data analysis and manuscript compilation. In that case, I'd like you to once more consider each aspect, in order to ensure that the data analysis and conclusions are balanced and comprehensive.

The obvious scientific highlight of the manuscript is the high temperature sensitivity of isoprene emissions from subarctic fen. The finding is rather exceptional and has potentially important implications, in so far as it requires and deserves extra careful consideration and special scrutiny. Regarding the analysis and conclusions reported in the manuscript, most of my concerns are either directly or indirectly related to this temperature sensitivity finding.

Conducting any scientific measurements is always by nature imperfect. To analyze something like temperature response of VOC flux, one has to take into consideration at least five aspects: 1) VOC flux per se, as a natural phenomenon, 2) Temperature per se, 3) Assumed relationship between T per se and VOC flux per se, 4) Measured estimate of VOC flux, and how complete or incomplete view that gives for VOC flux or emission per se, and finally, 5) Measured T, and its' ability to describe the thermal conditions relevant for VOC flux or emission. Failing with any of these aspects causes the risk that the analysis and conclusions on temperature response will be partly or fully misleading or incomplete.

I find no special need to believe that there would be any major shortcomings in the VOC flux measurement results of the manuscript. They also likely represent reality with

reasonable accuracy, so aspects #1 and #4 should be in order. Also, it can be quite safely assumed that T response follows some exponential form – that is so common feature in this field – so also the aspect #3 can be left above suspicion. That leaves us with aspects #2 and #5, T per se and measured T as an estimate of that.

The obvious merit of the manuscript is that it includes both air T and surface T as estimates for thermal conditions relevant for VOC flux. However, I suggest that a little bit more attention would be paid on discussing, how completely or incompletely these estimates are able to reflect the true thermal conditions controlling VOC (isoprene) sources. How sensitive these T measurements are to fail in producing accurate and precise description on relevant thermal conditions? Please note, that for example figure 5 gives the impression that if there is rather minor tendency to underestimate high surface temperatures and overestimate low surface temperatures, that would likely have a major impact on detected T response. I have no specific reason to believe that there would be such tendencies for under-/overestimation, but it's especially important to take even the slightest chance for those into account when exceptional T sensitivity is proposed.

In the manuscript, G93 emission model is applied for analyzing the isoprene fluxes. The core assumption is fairly expressed: it is assumed that the fen vegetation is a single big leaf. This assumption is of course inevitable, because there is lack of any other proper way to conduct model analysis on such flux data. However, it includes also significant limitations, mainly because of the inability of unshaded PAR measurement and single T measurement to reflect the true thermal and light conditions controlling VOC sources. Again, these limitations doesn't mean that one shouldn't use the model approach; one should just keep in mind that these limitations may complicate drawing the conclusions.

Although the fen surface vegetation is structurally very simple when compared to for example forest canopy, it still includes self-shading, causing full continuum from full light to totally shaded parts within the shallow layer of vegetation. This degree of self-shading

varies significantly depending on the angle of incoming radiation and/or proportion of diffuse radiation on total radiation. Therefore, any above-surface PAR measurement is judged to fail in fully describing the light conditions within the surface vegetation layer. Similarly, there is T gradient within the vegetation layer, which can't be properly described by air T measurement or even surface T measurement. Again, I have no specific reason to believe that your results would be untypically imprecise or inaccurate in this sense. Based on results presented in the manuscript, it's more and more obvious that models like G93-approach, especially CLxCT as such but also any form of T dependency, works best with simple structures, such as tree leaves. However, when they are applied for more complex structures such as surface vegetation layer, they struggle if the gradients in light and T environment are not described in detail. It's not that G93 or any other model or dependency description would be useless when used with 'single big leaf assumption' in case of complex structures, it's just that special caution is required when relevant findings are tried to distinguish among the features of model, assumptions and incompleteness of recorded environmental drivers. This is not stated as a challenge for the manuscript, but it's more to encourage you to once more consider, which of the conclusions based on findings and features presented for example in figures 5 and S5 are well-founded, given the limitations of 'single big leaf assumption' with complex vegetation structure.

From results and discussion as well as from figure S5 one gets the impression that light is not the primary control of fen isoprene fluxes. However, based on figures 2 and 3 there is remarkable similarity between PAR and fen isoprene flux. Or to be exact, at first glance there is similarity, but with a closer look there is also distinctive differences. First, before noon there is little variation in isoprene flux whereas during afternoon hours there is more variation. There is no similar bimodality in PAR recordings, but in air T there is some bimodality (figure 2), so it's obvious that thermal conditions have some effect on this difference between the different degree of variation in isoprene flux during morning hours and afternoons. Also, during morning hours PAR typically reaches value of 400 $\mu$mol m-2 s-1 soon after 6 o'clock in morning (which I believe should correspond

to CL=0.8, after which the isoprene fluxes tend to differ from zero, figure S5), but the detected isoprene fluxes in figure 3 differ from zero only some hours later. Overall, it would be very interesting to understand why the dynamics in fen isoprene fluxes differ between morning hours and afternoons. Would there be any chance to deepen the analysis in this sense? Also, do you have any guess why PAR=400 $\mu$mol m-2 s-1 looks like a threshold value for fen isoprene flux? What does that value for detected PAR would mean in regard to light conditions within the surface vegetation layer?

Detailed comments:

L34-56: This paragraph has all the right ingredients, but I felt that it's somehow difficult to follow. Please consider at least dividing it to two or more paragraphs (for example cut it from L45); or any other way to make this paragraph just a bit more easier to follow.

L126: Use of the term 'teflon' here and elsewhere in the manuscript. Please consider if the term could be removed. Originally, it's commercial brand-name. It's scientifically inexact, because nowadays it can refer to practically any fluoropolymer. And the exact name of your tube material is already mentioned in the same sentence.

L126: Please mention inner diameter instead of outer diameter, because the inner diameter is more relevant for the context.

L168: Was the T/RH probe equipped with radiation shield? Was it passively or actively ventilated? These details matter when one tries to judge how well the recorded T represents air T per se.

L230: This sentence is missing context. Emission from where? Please rephrase to clarify the context.

L254-255: Would there be difference in these Q10 values between morning hours and afternoon hours? Based on figure 3, there may be different type of response to environmental conditions between morning hours and afternoon hours. I understand that the lack of data points having high PAR somewhat limits partitioning this dataset

between morning hours and afternoon, but what if you include also data with PAR just below 1000 $\mu$mol m-2 s-1, let's say starting from 900 or 800 $\mu$mol m-2 s-1? Using PAR=800. . . . . .1000 $\mu$mol m-2 s-1 shouldn't be a problem, because at 800 $\mu$mol m-2 s-1 the CL is already > 0.96. These are suggestions that not necessarily lead to any reasonable result, but in case you haven't considered them earlier, they would potentially be interesting. Or what do you think, wouldn't it be even more unexpected if there is difference in T sensitivity between morning hours and afternoon?

L256-273: In this paragraph some very interesting points are raised. Could you consider to give recommendations for the community considering this issue? Maybe underline in the conclusions how important it is to measure T accurately, precisely and with such methods that it represents the true thermal conditions controlling the VOC production and release processes? The results and findings of the manuscript clearly support this kind of remark. Or should this be a topic for a follow-up article?

L294: I have nothing to contest; it's clear that biomass growth is a classical source of methanol. However, according to the figure S1, leaf expansion period has already ended before the mid-July measurement period. This methanol flux after the leaf expansion period is not surprising, because based on my own experience with boreal forest methanol emission continues throughout growing season (for example Aalto et al., 2014). I'd like to see more observations about cases when methanol emission is not clearly linked with biomass growth; here you would have chance to present one such case. Otherwise the community will keep repeating this Hüve et al. (2007) finding as the only relevant source for methanol emissions for another 13 years. I believe it's true finding but I also believe that it's not the whole truth.

L330-340: Were the depositions rates tested against relative humidity? At least in case of acetaldehyde it could be interesting. It's very likely that the potential effect of RH on these deposition rates returns to T (due to autocorrelation of T and RH), but there are chances that it could be vice versa: high humidity could be the main driver, instead of T itself. I don't ask you to add anything regarding RH, but just to consider by yourselves,

would accounting RH make sense for the analysis.

References: Hüve, K., Christ, M. M., Kleist, E., Uerlings, R., Niinemets, Ü., Walter, A. and Wildt, J.: Simultaneous growth and emission measurements demonstrate an interactive control of methanol release by leaf expansion and stomata, J. Exp. Bot., 58(7), 1783– 1793, doi:10.1093/jxb/erm038, 2007.

Aalto, J., Kolari, P., Hari, P., Kerminen, V.-M., Aaltonen, H., Levula, J., Siivola, E., Kulmala, M. and Bäck., J.: New foliage growth is a significant, unaccounted source for volatiles in boreal evergreen forests, Biogeosciences, 11, 1331-1344, doi:10.5194/bg-11-1331-2014, 2014.

---

## Referee Comment (RC2) · Anonymous Referee #2 · 19 Aug 2020

In their manuscript, "Volatile Organic Compound fluxes in a subarctic peatland and lake," Seco et al. present the results of flux measurements of volatile organic compounds (VOCs) at a subarctic fen and lake. The methods used are sound and are explained clearly and thoroughly. The results are important in that they provide one of the few measurements of VOC fluxes from these types of biomes in an understudied geographical region. The observations show that the fen is a source of many VOCs, particularly isoprene, and that the isoprene temperature response is stronger than is often assumed based on lower latitude data. Conversely, the lake appeared to be a sink of acetone and acetaldehyde. Overall, the study is of high quality and I recommend publication following minor revisions as described below.

As the authors point out, the results show that (1) there is a large difference between

[Figure]

the air temperature and vegetation surface temperature and (2) VOC emissions are extremely sensitive to temperature in this region. This is an important finding that can be used to improve model estimates of VOC emissions at high latitudes. It also suggests the importance of accurately measuring the temperature. I would suggest that the authors discuss in more detail the uncertainties associated with the temperature measurement method, how it compares with contact measurements of vegetation surfaces (or cite appropriate references). They should also discuss the uncertainty introduced by using a surface temperature measurement obtained some distance from the flux measurement site. Given these uncertainties, what is the uncertainty in the calculated Q10 values?

Lines 266-269: "Indeed, the response of our isoprene emissions to air temperature was even steeper (Q10 = 131; blue triangles in Fig. 5) than to surface temperature, which could translate into increased modelled isoprene emissions if implemented in models that do not calculate the vegetation temperature but instead use air temperature to drive biogenic VOC emissions." This sentence is overly long and while I understand what the authors are trying to say, it's not stated very clearly. I suggest separating into two sentences (replace the comma with a period), and rephrasing the second part of the existing sentence. In particular, the authors should more specifically state how the implementation of the Q10 result in models would lead to increased modelled isoprene emissions. It seems like an error would arise if there was a mismatch between the Q10 value and the temperatures used (i.e., using the high Q10 from air temp, but using leaf surface temps in the model, or vice versa) and the direction of the error would depend on the sense of the mismatch (Q10(air) + Tsurf vs. Q10(surf) + Tair). Please restate to improve clarity.

Lines 335-337: "Air temperature also influenced the flux of these two carbonyl VOCs. Acetone deposition was more intense at higher air temperatures, in July, when its mixing ratios were also higher (Fig. 6)." The authors state that air temperature "influenced the flux of these two carbonyl VOCs," but it seems like this could be simply correlation

rather than causation. The difference in flux between the two time periods (July and September) happens to coincide with a change in temperature, but also with a difference in mixing ratios. Also, the sense of the relationship is opposite in the two cases, with acetone deposition being higher at higher T, whereas acetaldehyde deposition is higher at lower T. Is there a mechanistic explanation for this? If not, and the relationship with temperature may not be causal, I would suggest rewording to clarify this.

Lines 357-358: "Instead, average methanol fluxes showed both net deposition and emission along the day during both seasons." How statistically significant is this conclusion? From Figure 3, it appears that the blue shaded region representing +/- 1 standard deviation overlaps 0 for most if not all data points (possibly excepting the last July data point, which I believe represents only a single measurement for that time window). Given the relatively sparse data and indicated standard deviations, wouldn't it be more accurate to say that the results indicate little to no flux (emission or uptake) of methanol to/from the lake?

Lines 395-396: "…similar to our July average of 4.7 $\pm$ 3.1 $\mu$mol m-2 day-1 (Table 1)." Referring back to lines 318-320, which state, "In particular, compounds such as DMS and monoterpenes had mean daily fluxes dominated by one or two hourly average data points that were not actually hourly averages, since they were based on only one measurement during that hour (i.e. data points without shading in Fig. 3)." along with the data shown in Figure 3, there appear to be two time periods with large positive fluxes representing single measurements. How were the daily averages calculated? Were they an average of all measurements equally weighted, or an average of the hourly averages? If the latter, that would give disproportionate weight to the high "hourly averages" that represent single data points. Please clarify in the text.

My last few comments regarding the lake fluxes suggest that the authors should consider alternate ways of analyzing and presenting the lake data to increase the statistical robustness of the results. For example, instead of hourly averages, they could consider averaging over 2 or 3 hour time periods to increase the number of data points in each

time period and improve the statistics, perhaps allowing for more definitive conclusions.

Minor grammatical changes:

Lines 270-273: "At the same Stordalen wetland as our study, Holst et al. (2010) found a steep temperature response to air temperatures above 15 °C, in agreement with our results (Fig. 5). Nevertheless, our high Q10 values corroborate that Arctic vegetation can have a stronger temperature sensitivity compared to plants from lower latitudes, which underpinned the most used biogenic emission models (Guenther et al., 2006), as already suggested from previous high-latitude studies (Holst et al., 2010; Kramshøj et al., 2016; Lindwall et al., 2016b, 2016a; Rinnan et al., 2014)." The second sentence is overly complicated. It obscures the important point that the high Q10 found in this study differs from model values based on lower latitudes. I would suggest rearranging the two thoughts into different sentences. Also, "underpinned" should be "underpin". E.g., "At the same Stordalen wetland as our study, Holst et al. (2010) found a steep temperature response to air temperatures above 15 °C, in agreement with our results (Fig. 5) and other high-latitude studies (Holst et al., 2010; Kramshøj et al., 2016; Lindwall et al., 2016b, 2016a; Rinnan et al., 2014). Our high Q10 values corroborate that Arctic vegetation can have a stronger temperature sensitivity compared to plants from lower latitudes, which underpin the most used biogenic emission models (Guenther et al., 2006)."

Lines 364-365: "Nevertheless, our available data showed maximum hourly average net emissions of 1 nmol m-2 s-1, being the daily average net rate of 0.24 ± 0.12 nmol m-2 s-1 (equivalent to 20 ± 10 $\mu$mol m-2 day-1; Table 1)." The wording of this sentence is awkward and confusing. Assuming I'm interpreting the authors' intent correctly, I would suggest replacing "being" with "and", e.g., "Nevertheless, our available data showed maximum hourly average net emissions of 1 nmol m-2 s-1, and a daily average net rate of 0.24 ± 0.12 nmol m-2 s-1 (equivalent to 20 ± 10 $\mu$mol m-2 day-1; Table 1)."

[Figure]

2020.

---

## Author Comment (AC1) · 17 Sep 2020

RESPONSE TO THE REVIEWERS  (ACP-2020-595)

We thank the reviewers for their encouraging comments and their careful reading of the manuscript. We have addressed each one of the points raised, and have altered text, figures and tables wherever appropriate. We look forward to having the editor's approval to submit a revised manuscript to be considered for final publication in ACP.

The referees' comments followed by our responses (R) are below.

**Juho Aalto (Referee #1)**

Seco et al. provide high-quality measurement data representing very relevant study object, VOC emissions from subarctic peatland and lake. Given how rare such datasets are, the results are obviously worth reporting, though the shortness and temporal limitedness of the data set apparent limitations concerning the data analysis and conclusions.These limitations should not be considered as weaknesses of the manuscript – they are more common features that should be taken into account, than any reason for rejection.

Overall, the manuscript fulfills the review criteria of ACP. The scientific question is well within the scope of ACP, both the data and conclusions are mainly valid, as well as the measurement and data analysis methods. The manuscript itself is mostly well structured and clearly expressed. There are some minor issues in the manuscript that may need to be reconsidered; those issues will be listed in my detailed comments.

Within the following section, I'll go through my main concerns regarding the data analysis and conclusions. Please note that many of my comments are more suggestions and questions aiming to clarify the results, than sound challenging based on solid scientific evidence. Chances are that many of the aspects pointed out by me have already been taken into consideration during the data analysis and manuscript compilation. In that case, I'd like you to once more consider each aspect, in order to ensure that the data analysis and conclusions are balanced and comprehensive.

R: We are glad that the referee considers our dataset and manuscript worthy of publication and thank him for his time in reviewing our work.

The obvious scientific highlight of the manuscript is the high temperature sensitivity of isoprene emissions from subarctic fen. The finding is rather exceptional and has potentially important implications, in so far as it requires and deserves extra careful consideration and special scrutiny. Regarding the analysis and conclusions reported

in the manuscript, most of my concerns are either directly or indirectly related to this temperature sensitivity finding.

Conducting any scientific measurements is always by nature imperfect. To analyze something like temperature response of VOC flux, one has to take into consideration at least five aspects: 1) VOC flux per se, as a natural phenomenon, 2) Temperature per se, 3) Assumed relationship between T per se and VOC flux per se, 4) Measured estimate of VOC flux, and how complete or incomplete view that gives for VOC flux or emission per se, and finally, 5) Measured T, and its' ability to describe the thermal conditions relevant for VOC flux or emission. Failing with any of these aspects causes the risk that the analysis and conclusions on temperature response will be partly or fully misleading or incomplete.

I find no special need to believe that there would be any major shortcomings in the VOC flux measurement results of the manuscript. They also likely represent reality with reasonable accuracy, so aspects #1 and #4 should be in order. Also, it can be quite safely assumed that T response follows some exponential form – that is so common feature in this field – so also the aspect #3 can be left above suspicion. That leaves us with aspects #2 and #5, T per se and measured T as an estimate of that.

The obvious merit of the manuscript is that it includes both air T and surface T as estimates for thermal conditions relevant for VOC flux. However, I suggest that a little bit more attention would be paid on discussing, how completely or incompletely these estimates are able to reflect the true thermal conditions controlling VOC (isoprene) sources. How sensitive these T measurements are to fail in producing accurate and precise description on relevant thermal conditions? Please note, that for example figure 5 gives the impression that if there is rather minor tendency to underestimate high surface temperatures and overestimate low surface temperatures, that would likely have a major impact on detected T response. I have no specific reason to believe that there would be such tendencies for under-/overestimation, but it's especially important to take even the slightest chance for those into account when exceptional T sensitivity is proposed.

R: It is true that a minor tendency to underestimate the high surface temperature and overestimate low surface temperatures could change the $Q_{10}$ of the temperature response. It is also true that our surface temperature readings were taken at the ICOS station, several hundred meters from the footprint of our measurements, but that is the only (and therefore, the best) vegetation surface temperature dataset we could obtain. We did measure with thermal cameras closer to our EC tower but, unfortunately, they were not functioning during the short time span when the BVOC fluxes were

functioning. Furthermore, it seems reasonable that the diurnal profile of surface temperature at the ICOS station would not differ much from the temperatures in our footprint area, located within the same mire complex. With that said, we also agree with the reviewer in that there is no reason to believe that the surface temperatures would have such tendencies of under- and overestimation.

In the manuscript, G93 emission model is applied for analyzing the isoprene fluxes. The core assumption is fairly expressed: it is assumed that the fen vegetation is a single big leaf. This assumption is of course inevitable, because there is lack of any other proper way to conduct model analysis on such flux data. However, it includes also significant limitations, mainly because of the inability of unshaded PAR measurement and single T measurement to reflect the true thermal and light conditions controlling VOC sources. Again, these limitations doesn't mean that one shouldn't use the model approach; one should just keep in mind that these limitations may complicate drawing the conclusions.
Although the fen surface vegetation is structurally very simple when compared to for example forest canopy, it still includes self-shading, causing full continuum from full light to totally shaded parts within the shallow layer of vegetation. This degree of self-shading varies significantly depending on the angle of incoming radiation and/or proportion of diffuse radiation on total radiation. Therefore, any above-surface PAR measurement is judged to fail in fully describing the light conditions within the surface vegetation layer. Similarly, there is T gradient within the vegetation layer, which can't be properly described by air T measurement or even surface T measurement. Again, I have no specific reason to believe that your results would be untypically imprecise or inaccurate in this sense. Based on results presented in the manuscript, it's more and more obvious that models like G93-approach, especially CLxCT as such but also any form of T dependency, works best with simple structures, such as tree leaves. However, when they are applied for more complex structures such as surface vegetation layer, they struggle if the gradients in light and T environment are not described in detail. It's not that G93 or any other model or dependency description would be useless when used with 'single big leaf assumption' in case of complex structures, it's just that special caution is required when relevant findings are tried to distinguish among the features of model, assumptions and incompleteness of recorded environmental drivers. This is not stated as a challenge for the manuscript, but it's more to encourage you to once more consider, which of the conclusions based on findings and features presented for example in figures 5 and S5 are well-founded, given the limitations of 'single big leaf assumption' with complex vegetation structure.

R: A similar 'big-leaf' analysis with the G93 algorithms was published several years ago for a *Sphagnum*-dominated fen in Southern Finland (Haapanala et al 2006). In that case, the authors tried to run a modification of the G93 model by simulating different light penetration inside the 2.5 cm deep active moss carpet. They concluded that there was no substantial improvement and that it was better to use the original G93 algorithm. Of course, the canopy of our fen was taller and more open than a moss carpet, because it was dominated by tall graminoids, and thus more prone to light and temperature differences at different levels.

First, we decided to keep the G93 in its original, simple form, not only because it is simpler and easier to calculate (therefore it is arguably better because it requires less "specialized" input data) but also because it allows better comparison with other studies that used the same approach in other ecosystems, such as the abovementioned fen in Finland. Second, if the surface temperature (and PAR) values that we used represented only the highest temperatures (and PAR) of the canopy, found only at the very top of the canopy, then we could assume that the canopy as a whole would experience lower temperatures (and PAR) due to self-shading closer to the ground. However, the ecosystem-scale isoprene fluxes we measured already represented the mix of the fluxes of the different levels the whole canopy. Considering these facts in Fig. 5, the resulting modified figure would be that the ecosystem isoprene fluxes would remain in place but the corresponding leaf (surface) temperatures, or at least the highest temperatures, would become lower than they are currently shown. Thus, the temperature dependency of isoprene fluxes would either remain the same (with fluxes starting to rise at lower temperatures than currently shown) or be even steeper than currently shown in Fig. 5. Nevertheless, in this case, the vegetation surface temperature sensor measured the temperature averaged over its field of view, where there were sunlit and shaded regions. All these facts suggest that the high temperature response seems to be a robust finding. Of course, future research could try to disentangle the contribution of the different layers of the canopy to the ecosystem isoprene flux.

From results and discussion as well as from figure S5 one gets the impression that light is not the primary control of fen isoprene fluxes. However, based on figures 2 and 3 there is remarkable similarity between PAR and fen isoprene flux. Or to be exact, at first glance there is similarity, but with a closer look there is also distinctive differences. First, before noon there is little variation in isoprene flux whereas during afternoon hours there is more variation. There is no similar bimodality in PAR recordings, but in air T there is some bimodality (figure 2), so it's obvious that thermal conditions have some effect on this difference between the different degree of variation in isoprene flux during morning hours and afternoons. Also, during morning hours PAR typically reaches value of 400 µmol m-2 s-1 soon after 6 o'clock in morning (which I believe should correspond to CL=0.8, after which the isoprene fluxes tend to differ from zero, figure S5), but the detected isoprene fluxes in figure 3 differ from zero only some hours later. Overall, it would be very interesting to understand why the dynamics in fen isoprene fluxes differ between morning hours and afternoons. Would there be any chance to deepen the analysis in this sense? Also, do you have any guess why PAR=400 µmol m-2 s-1 looks like a threshold value for fen isoprene flux? What does that value for detected PAR would mean in regard to light conditions within the surface vegetation layer?

R: These are all interesting questions; unfortunately, the present dataset does not encourage partitioning (morning/afternoon) given the limited amount of data points available. Future datasets with higher data availability may be used to investigate these suggestions.

Detailed comments:
L34-56: This paragraph has all the right ingredients, but I felt that it's somehow difficult to follow. Please consider at least dividing it to two or more paragraphs (for example cut it from L45); or any other way to make this paragraph just a bit more easier to follow.

R: Following the suggestion, we have divided the paragraph into two, cutting from line 45.

L126: Use of the term 'teflon' here and elsewhere in the manuscript. Please consider if the term could be removed. Originally, it's commercial brand-name. It's scientifically inexact, because nowadays it can refer to practically any fluoropolymer. And the exact name of your tube material is already mentioned in the same sentence.

R: We have now removed the term 'teflon' in this sentence and replaced it with PFA in the following sentence, the only two places where it appeared in the original manuscript.

L126: Please mention inner diameter instead of outer diameter, because the inner diameter is more relevant for the context.

R: We have now added the internal diameter of the tube: 1/4''.

L168: Was the T/RH probe equipped with radiation shield? Was it passively or actively ventilated? These details matter when one tries to judge how well the recorded T represents air T per se.

R: The temperature probe was equipped with a radiation shield that was passively ventilated. We have now added this information to the manuscript.

L230: This sentence is missing context. Emission from where? Please rephrase to clarify the context.

R: We have now given context and rephrased. The new sentence reads:

> Research on isoprene emissions from plant leaves has established that the instantaneous isoprene emission rate depends on the short-term (seconds to hours) light and temperature conditions (Monson et al., 2012).

L254-255: Would there be difference in these Q10 values between morning hours and afternoon hours? Based on figure 3, there may be different type of response to environmental conditions between morning hours and afternoon hours. I understand that the lack of data points having high PAR somewhat limits partitioning this dataset between morning hours and afternoon, but what if you include also data with PAR just below 1000 µmol m-2 s-1, let's say starting from 900 or 800 µmol m-2 s-1? Using PAR=800. . . . . .1000 µmol m-2 s-1 shouldn't be a problem, because at 800 µmol m-2 s-1 the CL is already > 0.96. These are suggestions that not necessarily lead to any reasonable result, but in case you haven't considered them earlier, they would potentially be interesting. Or what do you think, wouldn't it be even more unexpected if there is difference in T sensitivity between morning hours and afternoon?

R: Again, the morning/afternoon partitioning is an interesting question but this dataset is not the most appropriate for these kind of partitioning given the low number of available isoprene flux observations (half-hour data points with PAR ≥ 800 µmol m$^{-2}$ s$^{-1}$ and concurrent surface temperature readings were

only 35 and 31, respectively, for morning and afternoon). Nevertheless, it could very well be that there are differences along the day in the response to environmental variables, as some studies have found an influence of the circadian rhythm on isoprene emission fluxes (Hewitt et al, 2011). However, this needs more data for the analysis.

Following the reviewer suggestion, we have tested the temperature response with data points measured at PAR ≥ 800 µmol m$^{-2}$ s$^{-1}$, and included the following discussion in the appropriate section of the manuscript:

> For our calculation, we used only emission data points for which light was not a limiting factor (i.e. PAR $\geq 1000$ µmol m$^{-2}$ s$^{-1}$) to avoid interference from the correlation between light and temperature when fitting the temperature response. The $Q_{10}$ of isoprene emission in response to vegetation surface temperature was 14.5 and the $Q_{10}$ in response to air temperature was much higher, 131 (Fig. 5). Including in our calculation emission data points measured at PAR $\geq 800$ µmol m$^{-2}$ s$^{-1}$ increased the number of data points involved (from 52 to 66 at PAR $\geq 1000$ and $\geq 800$ µmol m$^{-2}$ s$^{-1}$, respectively) at the expense of potential light-response interference, although according to the G93 model the light-response effect would be small ($C_L > 0.96$ at PAR=800 µmol m$^{-2}$ s$^{-1}$). The lower-PAR $Q_{10}$ for the surface temperature was 10.3, still above the range 3-6 of the G93 and other biogenic models. The $Q_{10}$ for the air temperature was 18.6, much lower than for the higher PAR, probably due to the less direct relationship of isoprene emissions with air temperature. Altogether, these calculations suggest a robust, strong response of isoprene fluxes to temperature.

L256-273: In this paragraph some very interesting points are raised. Could you consider to give recommendations for the community considering this issue? Maybe underline in the conclusions how important it is to measure T accurately, precisely and with such methods that it represents the true thermal conditions controlling the VOC production and release processes? The results and findings of the manuscript clearly support this kind of remark. Or should this be a topic for a follow-up article?

R: This is an important point and thus we have included the following text in the "concluding remarks":

> Our measurements also displayed the disparity between the temperature of the air and that of the vegetation surface, the latter being several degrees warmer during daytime (Figs. 2 and 5). Consequently, it is advisable that future VOC studies measure accurately and precisely the vegetation temperatures that represent the thermal conditions controlling the VOC production and release processes. Furthermore, while we do not suggest taking these $Q_{10}$ values as true coefficients to be directly implemented into modelling, it is worth mentioning that care should be taken when applying $Q_{10}$ values in models. Otherwise, a mismatch could translate into erroneous results, for instance when using a $Q_{10}$ derived from the response to air temperature in models that drive VOC emissions with leaf temperature and vice versa.

L294: I have nothing to contest; it's clear that biomass growth is a classical source of methanol. However, according to the figure S1, leaf expansion period has already ended before the mid-July measurement period. This methanol flux after the leaf expansion period is not surprising, because based on my own experience with boreal forest methanol emission continues throughout growing season (for example Aalto et al., 2014). I'd like to see more observations about cases when methanol emission is not clearly linked with biomass growth; here you would have chance to present one such case. Otherwise the community will keep repeating this Hüve et al. (2007) finding as the only relevant source for methanol emissions for another 13 years. I believe it's true finding but I also believe that it's not the whole truth.

R: We agree that methanol is emitted throughout the growing season. Indeed, our July measurements correspond to a period when leaf expansion was not at its seasonal strongest time. We showed in Fig. 3 that methanol emission fluxes in July described a diurnal cycle, and we commented (lines 280-281) also that Holst et al (2010) reported methanol fluxes at the same fen several years ago also outside of the peak in leaf expansion (in August, even later than our measurements). Furthermore, we mentioned (L285-287) a reference to other studies of methanol fluxes that are not restricted to the leaf expansion (Wohlfahrt et al 2015). Therefore, in our original manuscript, we did not imply that the only source of methanol is biomass growth. That paragraph gave a hint to the reader of the multiple and complex controls on the emissions of these non-isoprenoid VOCs, other than light and temperature. Nevertheless, it is true that methanol is typically emitted in higher amounts during leaf elongation, as shown for boreal forests in Aalto et al (2014) as well.

We have now expanded that section, which now reads:

Methanol can be emitted constitutively by plants throughout their growing season, with increased release linked to leaf expansion (Aalto et al., 2014; Hüve et al., 2007) and emission bursts elicited by herbivore feeding (Peñuelas et al., 2005a). Methanol emissions have also been associated with soils. For example, methanol was one of the main compounds released from subalpine forest floor (Gray et al., 2014) and thawing permafrost (Kramshøj et al., 2018).

L330-340: Were the depositions rates tested against relative humidity? At least in case of acetaldehyde it could be interesting. It's very likely that the potential effect of RH on these deposition rates returns to T (due to autocorrelation of T and RH), but there are chances that it could be vice versa: high humidity could be the main driver, instead of T itself. I don't ask you to add anything regarding RH, but just to consider by yourselves, would accounting RH make sense for the analysis.

R: We have checked the relationship of RH with the deposition rates of these carbonyl compounds. In the case of acetone, the relationship resembled very much that of temperature, with higher deposition rates matching the lower RH values expected for those higher temperatures. Thus, higher RH did not drive deposition of acetone. Acetaldehyde deposition, just like with temperature, did not show a clear relationship with RH. The strongest deposition rates were broadly scattered around 70-90% RH but with most of the near-zero fluxes at around RH of 85-100%, so again there was no clear relationship to RH.

References:
Hüve, K., Christ, M. M., Kleist, E., Uerlings, R., Niinemets, Ü., Walter, A. and Wildt, J.: Simultaneous growth and emission measurements demonstrate an interactive control of methanol release by leaf expansion and stomata, J. Exp. Bot., 58(7), 1783– 1793, doi:10.1093/jxb/erm038, 2007.
Aalto, J., Kolari, P., Hari, P., Kerminen, V.-M., Aaltonen, H., Levula, J., Siivola, E., Kulmala, M. and Bäck., J.: New foliage growth is a signiïn˜A̧cant, unaccounted source for volatiles in boreal evergreen forests, Biogeosciences, 11, 1331-1344, doi:10.5194/bg-11-1331-2014, 2014.

**Anonymous Referee #2**

In their manuscript, "Volatile Organic Compound fluxes in a subarctic peatland and lake," Seco et al. present the results of flux measurements of volatile organic compounds (VOCs) at a subarctic fen and lake. The methods used are sound and are explained clearly and thoroughly. The results are important in that they provide one of the few measurements of VOC fluxes from these types of biomes in an understudied geographical region. The observations show that the fen is a source of many VOCs, particularly isoprene, and that the isoprene temperature response is stronger than is often assumed based on lower latitude data. Conversely, the lake appeared to be a sink of acetone and acetaldehyde. Overall, the study is of high quality and I recommend publication following minor revisions as described below.

R: We thank the referee for his/her effort in reviewing and are glad of his/her positive opinion of the manuscript.

As the authors point out, the results show that (1) there is a large difference between the air temperature and vegetation surface temperature and (2) VOC emissions are extremely sensitive to temperature in this region. This is an important finding that can be used to improve model estimates of VOC emissions at high latitudes. It also suggests the importance of accurately measuring the temperature. I would suggest that the authors discuss in more detail the uncertainties associated with the temperature measurement method, how it compares with contact measurements of vegetation surfaces (or cite appropriate references). They should also discuss the uncertainty introduced by using a surface temperature measurement obtained some distance from the flux measurement site. Given these uncertainties, what is the uncertainty in the calculated Q10 values?

R: We agree that it is important to accurately measure the temperature of the ecosystem that we are studying. That is why we have included a reminder in the "concluding remarks" section of the article (see response to reviewer #1 regarding lines L256-273).

We have also added more information on the measurement of the vegetation surface temperature in the "methods" section of the manuscript, which now reads:

We used the vegetation surface temperature, retrieved with an infrared radiometer (SI-111, Apogee Instruments, Logan UT, USA), from the nearby ICOS (Integrated Carbon Observation System) Sweden measurements within the same Stordalen Mire complex (Fig. 1). A technology used for

decades (Fuchs and Tanner, 1966), the infrared radiometer measured non-invasively the vegetation surface temperature integrated over its field of view (4.8 m$^2$). Even though these radiation readings took place a couple of hundred meters away from our flux footprint, it was the best available proxy of the temperature experienced by the vegetation surface of the fen. Indeed, previous studies have shown that solar radiation can warm Arctic plants several degrees above their surrounding air temperature (Lindwall et al., 2016a; Wilson, 1957).

Regarding the uncertainty of the $Q_{10}$ values, assuming that the measured temperature (either air or surface temperature) has a precision of 1% –as indicated by the sensor manufacturers– and that it is accurate, then most of the uncertainty in the $Q_{10}$ coefficient comes from the isoprene mixing ratios – which have 15% uncertainty. We have now also calculated the $Q_{10}$ values using more data than we did in the first version of the manuscript (see our answer to reviewer #1 regarding lines L254-255), which illustrates that our $Q_{10}$ coefficients shows some degree of variability, but they always corroborate a higher sensibility to temperature than current emission models. Lastly, we must add that our study was not designed to determine the $Q_{10}$ values as such and that they should not be taken as an absolute truth. Instead, we use the $Q_{10}$ values to highlight the strong temperature dependence of our fen.

Lines 266-269: "Indeed, the response of our isoprene emissions to air temperature was even steeper (Q10 = 131; blue triangles in Fig. 5) than to surface temperature, which could translate into increased modelled isoprene emissions if implemented in models that do not calculate the vegetation temperature but instead use air temperature to drive biogenic VOC emissions." This sentence is overly long and while I understand what the authors are trying to say, it's not stated very clearly. I suggest separating into two sentences (replace the comma with a period), and rephrasing the second part of the existing sentence. In particular, the authors should more specifically state how the implementation of the Q10 result in models would lead to increased modelled isoprene emissions. It seems like an error would arise if there was a mismatch between the Q10 value and the temperatures used (i.e., using the high Q10 from air temp, but using leaf surface temps in the model, or vice versa) and the direction of the error would depend on the sense of the mismatch (Q10(air) + Tsurf vs. Q10(surf) + Tair). Please restate to improve clarity.

R: We agree that this sentence is not well stated. Now, we have removed the second part from this section of the manuscript and rewritten the main idea in the "concluding remarks" (see our answer to reviewer #1 regarding lines L256-273).

Lines 335-337: "Air temperature also influenced the flux of these two carbonyl VOCs. Acetone deposition was more intense at higher air temperatures, in July, when its mixing ratios were also higher (Fig. 6)." The authors state that air temperature "influenced the flux of these two carbonyl VOCs," but it seems like this could be simply correlation rather than causation. The difference in flux between the two time periods (July and September) happens to coincide with a change in temperature, but also with a difference in mixing ratios. Also, the sense of the relationship is opposite in the two cases, with acetone deposition being higher at higher T, whereas acetaldehyde deposition is higher at lower T. Is there a mechanistic explanation for this? If not, and the relationship with temperature may not be causal, I would suggest rewording to clarify this.

R: We agree that the text needs to be clarified. Our intention was just to state the relationship with temperature observed from the plots in Fig. 6. It is known that atmospheric mixing ratios can drive the flux direction to/from water of these water soluble short-chain oxygenated VOCs, and the relationship with temperature may just be a coincidence. We have now reworded this paragraph; this is the final text:

There was a correlation of the acetaldehyde and acetone deposition rates with their corresponding atmospheric mixing ratios (Fig. 6), with increasing deposition at higher mixing ratios, resulting in average deposition velocities of -0.23 ± 0.01 and -0.68 ± 0.03 cm s$^{-1}$ for acetone and acetaldehyde, respectively. The high water solubility of these short-chain oxygenated VOCs helps their deposition from the air to the water, and may partly explain the correlation of the deposition rate with their atmospheric mixing ratios (Fig. 6). The flux of these two carbonyl VOCs showed a relationship to air temperature as well, that might very well be coincidental and dependent on atmospheric mixing ratios. Acetone deposition was more intense at higher air temperatures, in July, when its mixing ratios were also higher (Fig. 6). In contrast to acetone, acetaldehyde did not present a clear relationship with air temperature but its strongest deposition rates occurred at air temperatures below 3 ℃, concurrent with higher mixing ratios (Fig. 6).

Lines 357-358: "Instead, average methanol fluxes showed both net deposition and emission along the day during both seasons." How statistically significant is this conclusion? From Figure 3, it appears that the blue shaded region representing +/- 1 standard deviation overlaps 0 for most if not all data points (possibly excepting the last July data point, which I believe represents only a single measurement for that time window). Given the relatively sparse data and indicated standard deviations, wouldn't it be more accurate to say that the results indicate little to no flux (emission or uptake) of methanol to/from the lake?

R: We concur with the reviewer and have modified that sentence to:

> Instead, average methanol fluxes showed little to no flux, either deposition or emission, along the day during both seasons.

Lines 395-396: ". . .similar to our July average of 4.7 ± 3.1 µmol m-2 day-1 (Table 1)." Referring back to lines 318-320, which state, "In particular, compounds such as DMS and monoterpenes had mean daily fluxes dominated by one or two hourly average data points that were not actually hourly averages, since they were based on only one measurement during that hour (i.e. data points without shading in Fig. 3)." along with the data shown in Figure 3, there appear to be two time periods with large positive fluxes representing single measurements. How were the daily averages calculated? Were they an average of all measurements equally weighted, or an average of the hourly averages? If the latter, that would give disproportionate weight to the high "hourly averages" that represent single data points. Please clarify in the text.

R: The daily averages were calculated as an average of the hourly averages. The reason is that, for each VOC species, the number of available data points at each hour was variable among the 24 hours of the day. Giving an equal weight to each data point would give a disproportionate weigh to the values of the hours with more data, which tend to be the central parts of the day due to better turbulence conditions and clearer VOC fluxes, or those hours that saw more frequently the wind blowing from the right direction. The downside is that an "average" from a single data point that is disproportionately different from the other data points can distort the daily average. This is just what we discussed in lines 318-320 of the preprint. We have now clarified the calculation of the daily averages by including the following text in the caption of Table 1:

> These daily averages were calculated from the hourly averages shown in Figs. 3 and 4.

My last few comments regarding the lake fluxes suggest that the authors should consider alternate ways of analyzing and presenting the lake data to increase the statistical robustness of the results. For example, instead of hourly averages, they could consider averaging over 2 or 3 hour time periods to increase the number of data points in each time period and improve the statistics, perhaps allowing for more definitive conclusions.

R: That is a good suggestion and we tried this approach with the lake monoterpenes and DMS in July. Averaging over 2 hours did only slightly change the trace shown in Fig.3 and the daily average was practically the same given the high uncertainty of the original daily averages shown in Table 1. This is understandable because there are few points measured during the hours surrounding the "outliers" too, so the effect of the "outliers" is still apparent in the new averages. Averaging over 3 hours made the daily pattern more constant (but with high standard deviation and of course less time resolution), which may better reflect the real flux behavior or not, because we do not know what is the real flux behavior.

Therefore, we decided to keep the original hourly averages for these two compounds, like all the other reported VOCs and environmental variables, instead of obscuring a few high values by averaging with neighboring data points. We maintained the same cautionary comment in the text (lines 318-320 in the original preprint), so the reader can have a better idea of the available data and their shortcomings.

Minor grammatical changes:
Lines 270-273: "At the same Stordalen wetland as our study, Holst et al. (2010) found a steep temperature response to air temperatures above 15 _C, in agreement with our results (Fig. 5). Nevertheless, our high Q10 values corroborate that Arctic vegetation can have a stronger temperature sensitivity compared to plants from lower latitudes, which underpinned the most used biogenic emission models (Guenther et al., 2006), as already suggested from previous high-latitude studies (Holst et al., 2010; Kramshøj et al., 2016; Lindwall et al., 2016b, 2016a; Rinnan et al., 2014)." The second sentence is overly complicated. It obscures the important point that the high Q10 found in this study differs from model values based on lower latitudes. I would suggest rearranging the two thoughts into different sentences. Also, "underpinned" should be "underpin". E.g., "At the

same Stordalen wetland as our study, Holst et al. (2010) found a steep temperature response to air temperatures above 15 _C, in agreement with our results (Fig. 5) and other high-latitude studies (Holst et al., 2010; Kramshøj et al., 2016; Lindwall et al., 2016b, 2016a; Rinnan et al., 2014). Our high Q10 values corroborate that Arctic vegetation can have a stronger temperature sensitivity compared to plants from lower latitudes, which underpin the most used biogenic emission models (Guenther et al., 2006)."

R: We have improved the readability of these sentences following the reviewer's advice.

Lines 364-365: "Nevertheless, our available data showed maximum hourly average net emissions of 1 nmol m-2 s-1, being the daily average net rate of 0.24 ± 0.12 nmol m-2 s-1 (equivalent to 20 ± 10 µmol m-2 day-1; Table 1)." The wording of this sentence is awkward and confusing. Assuming I'm interpreting the authors' intent correctly, I would suggest replacing "being" with "and", e.g., "Nevertheless, our available data showed maximum hourly average net emissions of 1 nmol m-2 s-1, and a daily average net rate of 0.24 ± 0.12 nmol m-2 s-1 (equivalent to 20 ± 10 µmol m-2 day-1; Table 1)."

R: We have improved the readability of these sentences following the reviewer's advice.

REFERENCES

Fuchs, M. and Tanner, C. B.: Infrared Thermometry of Vegetation, Agron. J., 58(6), 597–601, doi:10.2134/agronj1966.00021962005800060014x, 1966.

Gray, C. M., R. K. Monson, and N. Fierer (2014), Biotic and abiotic controls on biogenic volatile organic compound fluxes from a subalpine forest floor, J. Geophys. Res. Biogeosci., 119, 547–556, doi:10.1002/2013JG002575.

Haapanala, S., Rinne, J., Pystynen, K.-H., Hellén, H., Hakola, H. and Riutta, T.: Measurements of hydrocarbon emissions from a boreal fen using the REA technique, Biogeosciences, 3(1), 103–112, doi:10.5194/bg-3-103-2006, 2006.

Hewitt, C., Ashworth, K., Boynard, A. et al. Ground-level ozone influenced by circadian control of isoprene emissions. Nature Geosci 4, 671–674 (2011). https://doi.org/10.1038/ngeo1271

Holst, T., Arneth, A., Hayward, S., Ekberg, A., Mastepanov, M., Jackowicz-Korczynski, M., Friborg, T., Crill, P. M. and Backstrand, K.: BVOC ecosystem flux measurements at a high latitude wetland site, Atmos. Chem. Phys., 10(4), 1617–1634, 2010.

Kramshøj M., Albers C.N., Holst T., Holzinger R., Elberling B., Rinnan R. (2018) Biogenic volatile release from permafrost thaw is determined by the soil microbial sink. Nature Communications 9: 3412.

Lindwall, F., Schollert, M., Michelsen, A., Blok, D. and Rinnan, R.: Fourfold higher tundra volatile emissions due to arctic summer warming, J. Geophys. Res. Biogeosciences, 121(3), 895–902, doi:10.1002/2015JG003295, 2016a.

Peñuelas, J., Filella, I., Stefanescu, C. and Llusia, J.: Caterpillars of Euphydryas aurinia (Lepidoptera: Nymphalidae) feeding on Succisa pratensis leaves induce large foliar emissions of methanol, New Phytol., 167(3), 851–857, 2005a.

Wilson, J. W.: Observations on the Temperatures of Arctic Plants and Their Environment, J. Ecol., 45(2), 499, doi:10.2307/2256933, 1957.

Wohlfahrt, G., Amelynck, C., Ammann, C., Arneth, A., Bamberger, I., Goldstein, A. H., Gu, L., Guenther, A., Hansel, A., Heinesch, B., Holst, T., Hörtnagl, L., Karl, T., Laffineur, Q., Neftel, A., McKinney, K., Munger, J. W., Pallardy, S. G., Schade, G. W., Seco, R. and Schoon, N.: An ecosystem-scale perspective of the net land methanol flux: synthesis of micrometeorological flux measurements, Atmos. Chem. Phys., 15(13), 7413–7427, doi:10.5194/acp-15-7413-2015, 2015.